# Detecting genuine multipartite entanglement in multi-qubit devices with restricted measurements

Nicky Kai Hong Li [1,2,3] ✉, Xi Dai [4,5] ✉, Manuel H. Muñoz-Arias [6], Kevin Reuer [4,5], Marcus Huber [1,2,3] & Nicolai Friis [1,2] ✉

Detecting genuine multipartite entanglement (GME) is a state-characterization task that benchmarks coherence and experimental control in quantum systems. Existing GME tests often require joint measurements on many qubits, posing challenges for systems like time-bin encoded qubits and microwave photons from superconducting circuits, where qubit connectivity is limited or measurement noise grows with the number of jointly measured qubits. Here we introduce versatile GME and $k$-inseparability criteria applicable to any state, which only require measuring $O(n^2)$ out of $2^n$ (at most) $m$-body stabilizers of $n$-qubit target graph states, with $m$ upper-bounded by twice the graph's maximum degree. For cluster or ring-graph states, only constant-weight stabilizers are needed. Using semidefinite programming (and sometimes graph-local complementations), we can reduce the number or weight of required stabilizers. Analytical and numerical results show that our criteria are noise-robust and may infer state infidelity from certified $k$-inseparability in microwave photonic graph states generated under realistic conditions.

Current developments of quantum technologies have enabled the simultaneous control of increasingly large systems and the preparation of highly complex quantum states (see, e.g., refs. 1–4). These advances demand suitable techniques to characterize and benchmark the corresponding devices and the states that they produce. However, increasing system sizes mean that full state tomography quickly becomes infeasible, shifting the focus to scalable but less all-encompassing characterization tools that still capture genuine quantum features as well as the quality of system control. Useful alternatives typically relying on few measurements present themselves in techniques such as shadow tomography[5,6] and entanglement certification (see, e.g., ref. 7), which represent two ends of a spectrum: the former employs randomized/arbitrary measurements to create a classical representation of the state for broad *a posteriori* predictions, while the latter typically relies on targeted measurements to directly

extract information on quantum correlations between subsystems. Yet, even these specific measurements are often difficult to implement in practice, prompting purpose-designed solutions for entanglement certification across a variety of platforms, including, e.g., spatial[8,9] and temporal[10] degrees of freedom of photons, spin ensembles[11,12], and recently even setups not traditionally employed for quantum-information processing like electron-photon pairs in electron microscopy[13,14]. Under these circumstances, it becomes essential to extend the toolbox of entanglement-detection criteria to meet various platform-specific restrictions. Here, we provide such a practically implementable criterion for detecting *genuine multipartite entanglement* (GME) in systems that permit joint operations and measurements only on specific subsets of qubits.

Multipartite entanglement is a crucial resource for quantum technologies[15], enabling applications from quantum

[1]Technische Universität Wien, Atominstitut, Vienna, Austria. [2]Vienna Center for Quantum Science and Technology, TU Wien, Vienna, Austria. [3]Institute for Quantum Optics and Quantum Information (IQOQI), Austrian Academy of Sciences, Vienna, Austria. [4]Department of Physics, ETH Zurich, Zurich, Switzerland. [5]Quantum Center, ETH Zurich, Zurich, Switzerland. [6]Institut Quantique and Département de Physique, Université de Sherbrooke, Sherbrooke, Canada. ✉e-mail: kai.li@tuwien.ac.at; xi.dai@phys.ethz.ch; nicolai.friis@tuwien.ac.at

communication[16–19], measurement-based quantum computation (MBQC)[20,21], to quantum error correction[22], quantum metrology[23], and other quantum algorithms[24]. In particular, certain multipartite tasks require GME specifically[16,25,26]. But what is GME? It formalizes the idea that some multipartite states are entangled across every bipartition (a grouping of subsystems into two subsets), but can still be produced as statistical mixtures of states that have only bipartite entanglement across different bipartitions but no multipartite entanglement themselves. The latter are called *biseparable*, whereas states that cannot be decomposed into mixtures of bipartite entangled states are GME. This concept extends to the more general *k-(in)separability*, where entanglement across all *k*-partitions is considered.

The study of GME is a thriving subject of ongoing research. Since the seminal works on this topic[27–29], much progress has been made in developing GME-detection methods (see, e.g., the reviews[7,30,31]). The two crucial pieces of information for choosing a suitable detection strategy for a given setup are (i) the target states one expects to encounter, and (ii) which measurements can reasonably be carried out in the physical platform at hand. While the former influences how well a chosen GME criterion works in practice, the latter restricts which criteria one may evaluate in the first place. Consequently, the GME-detection toolbox is continuously being expanded to cover relevant platforms and target states.

In this work, we introduce a novel family of criteria designed to detect GME in the important class of *n*-qubit graph states (and states close to them), in scenarios where joint measurements can only be carried out on certain "local" subsets of connected qubits, with the locality and connectivity dictated by the associated graph structure. Such restrictions naturally arise in setups producing time-bin encoded qubits, e.g., time-multiplexed photons in the near-visible spectrum generated from spontaneous parametric down-conversion in combination with Sagnac interferometers[32], or pairs of optical parametric amplifiers and Mach-Zehnder interferometers[33–35]. There, joint measurements and operations are usually restricted to temporally adjacent photons. Our criteria are also well-suited for platforms where single-qubit measurement efficiency is limited. In such systems, the number of measurements required to estimate a weight-*m* observable scales exponentially with *m* and becomes experimentally prohibitive if *m* scales with the system size. This measurement-efficiency limitation is the main bottleneck encountered in characterizing microwave photons generated from superconducting circuits[36–39]. A similar, though less severe, limitation also appears in optical photonic platforms, where sequential arbitrary-basis single-qubit measurements rely on electro-optical modulators for fast basis switching, whose losses reduce the overall detection efficiency[40,41].

Crucially, such setups naturally generate graph states central to quantum information processing, including two-dimensional cluster states for MBQC[20,21], ring-graph states for fusion-based quantum computing[42], or tree-graph states for error correction[43] and constructing quantum repeaters[44]. Therefore, certifying multipartite entanglement of these states is motivated by the need to characterize resources in quantum information processing, and given the practical restrictions of the setups, producing these states becomes an imperative that makes the techniques presented here highly relevant for device benchmarking.

Numerous methods exist for detecting GME, including stabilizer-based witnesses[45–47], PPT-inspired criteria[48,49], fidelity bounds[50,51], as well as permutation inequalities and moments[52–55]. However, nearly all require measuring observables acting on up to *O*(*n*) qubits, which limits their experimental feasibility when only few-body measurements are accessible. Moreover, many detect only GME versus full separability, without resolving intermediate levels of *k*-inseparability (see Supplementary Note 1 for a review of pertinent previous methods). This motivates the approaches developed here, which are designed for scenarios restricted to *O*(1)-body measurements for important classes of graph states.

The entanglement criteria we introduce here are versatile yet simple, making them applicable across a broad range of physical platforms, as they only require measuring few stabilizers of chosen target graph states. Each stabilizer is a tensor product of Pauli operators acting on at most twice the graph's maximum degree, and the number of stabilizers to be measured scales at most quadratically with the number of vertices. Since all stabilizer states are local-unitarily equivalent to graph states[56,57], our criteria can certify multipartite entanglement—a generally NP-hard task[58,59]—for many relevant states in quantum information processing, particularly those close to stabilizer states. In addition, we show how our criteria can be used even in setups where only a subset of the few measurements mentioned above can be carried out by utilizing graph-local complementations and semidefinite programming (SDP).

We demonstrate the performance and versatility of our method in two ways. First, we evaluate the entanglement criteria for several pertinent graph states and non-stabilizer states that are local-unitary (LU) transformed Dicke states with added white noise, providing analytical noise thresholds for each case, with the latter examples demonstrating that our method applies beyond stabilizer states. Second, we showcase the applicability of our criteria for benchmarking quantum devices by numerically simulating microwave-photonic graph states generated from superconducting circuits[38] and quantitatively comparing our method to other approaches.

The remainder of this article is structured as follows. In section "Background and notation", we provide basic definitions for graph states and multipartite entanglement. In sections "GME & *k*-inseparability criteria" and "SDP for incomplete measurements", we present and discuss our new GME/*k*-inseparability criteria, as well as our fixed *k*-partition inseparability criteria, and demonstrate how SDP can be used to bound certain observables when direct measurements are not feasible. We then test our criteria's performance in a simulated experiment with microwave-photonic qubits in the remaining Results sections, showing that the certified *k*-inseparability may infer state infidelity. Finally, we provide a discussion and outlook.

## Results
### Background and notation
Let us first state some definitions from graph theory that are relevant for this paper. Let $G = (V, E)$ be an *n*-vertex graph with vertex set $V = \{1, ..., n\} = : [n]$ and (unweighted) edge set $E := \{(i, j)|i \in V, j \in N(i)\}$ where $N(i)$ denotes the neighborhood of vertex $i$. In this work, we only consider undirected graphs, so we identify $(i, j)$ and $(j, i)$ as the same, single element in $E$. A *k-partition* of a graph $G$ is a division of the vertex set $V$ into $k$ disjoint subsets. A *k-cut* is a set of edges whose removal results in $k$ connected subgraphs that are disconnected from each other. In the rest of the paper, we will interchangeably use the two terms to refer to a particular way to divide a graph into $k$ connected parts. A *matching* of a graph $G$ is a set of pairwise non-adjacent edges. A matching is *maximal* if it is not a subset of any other matching. A matching is *maximum-cardinality* if it contains the most edges of $G$ and is necessarily maximal.

For every graph $G$, the associated graph state $|G\rangle$ can be defined as $|G\rangle = \prod_{(i,j)\in E} CZ_{ij} |+\rangle^{\otimes n}$ where $CZ_{ij} = (|0\rangle \langle 0|_i \otimes \mathbb{1}_j + |1\rangle \langle 1|_i \otimes Z_j) \otimes \mathbb{1}_{[n]\setminus\{i,j\}} = CZ_{ji}$ is the controlled-$Z$ (or controlled-phase) gate on qubits $i$ and $j$. The stabilizer group Stab$(|G\rangle)$ of a graph state $|G\rangle$ is generated by the stabilizers $S_i := X_i \bigotimes_{j\in N(i)} Z_j$ of all vertices $i \in V$, which satisfy $S_i|G\rangle = |G\rangle$ and $[S_i, S_j] = 0 \; \forall \; i, j$.

Let us move on to state some relevant definitions related to multipartite entanglement. The state of $n$ quantum systems with associated (finite-dimensional) Hilbert spaces $\mathbb{C}^{d_i}$ with dimensions $d_i$ for $i = 1, 2, ..., n$ is described by a density matrix $\rho$, an element of the space $\mathcal{D}(\mathbb{C}^{d_1} \otimes ... \otimes \mathbb{C}^{d_n})$ of normalized (Tr($\rho$) = 1), positive semi-definite ($\rho \geqslant 0$) operators. A state $\rho$ is *k-separable* if and only if it can be expressed as a statistical mixture of density operators that factorize

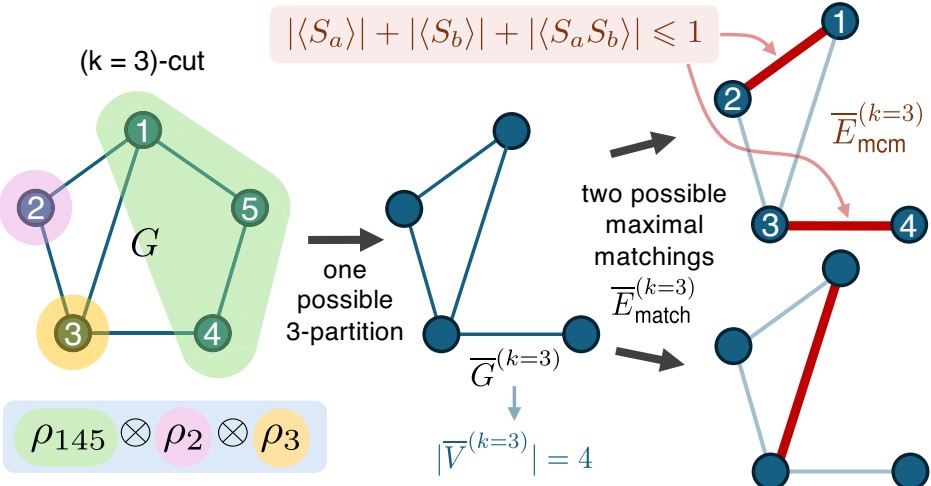

**Fig. 1 | Graphical illustration of what a $k$-partition subgraph $\overline{G}^{(k)}$ of the graph $G$, the quantity $|\overline{V}^{(k)}|$, and the maximum cardinality matching $\overline{E}_{\text{mcm}}^{(k)}$ of $\overline{G}^{(k)}$ represent.** In a 3-partition, we partition $G$ into three parts (highlighted in different colors). Computing $|\overline{V}^{(k)}|$ and $\overline{E}_{\text{mcm}}^{(k)}$ for, e.g., the color-labeled 3-partition corresponds to evaluating the separability bound for all states of the form $\rho_{145} \otimes \rho_2 \otimes \rho_3$. By removing edges that do not connect vertices in different regions—the two edges contained in the green region on the right-hand side of $G$—we obtain the subgraph $\overline{G}^{(k=3)}$. There are only two possible non-isomorphic maximal matchings $\overline{E}_{\text{match}}^{(k=3)}$ of $\overline{G}^{(k=3)}$, which are represented by the red edges. The top right matching has the most edges, making it the maximum-cardinality matching $\overline{E}_{\text{mcm}}^{(k=3)}$ of this particular 3-partition. This set corresponds to the maximum reduction in the analytic upper bound using the *anticommutativity inequality* for states that are separable across the partitions that cut through the edges in $\overline{E}_{\text{mcm}}^{(k=3)}$, each contributing to a reduction of 2 in the upper bound as $|\langle S_a \rangle| + |\langle S_b \rangle| + |\langle S_a S_b \rangle| \leq 1$ for $(a, b) \in \{(1, 2), (3, 4)\}$ (see Methods for details).

into a tensor product of density operators of $k$ subsystems that each comprise one or more of the original $n$ subsystems, formally

$$\rho = \sum_i p_i \left| \psi_i^{[A_1^i]} \right\rangle \left\langle \psi_i^{[A_1^i]} \right| \otimes \ldots \otimes \left| \psi_i^{[A_k^i]} \right\rangle \left\langle \psi_i^{[A_k^i]} \right|, \quad (1)$$

where for each $i$, the disjoint sets $A_1^i, \ldots, A_k^i$ (such that $\cup_{j=1}^k A_j^i = [n]$) denote a $k$-partition of the $n$ subsystems and $\left| \psi_i^{[A_j^i]} \right\rangle \in \bigotimes_{\alpha \in A_j^i} \mathbb{C}^{d_\alpha}$. We call a state *genuinely multipartite entangled* (GME) if it is not biseparable (2-separable), while states of $n$ subsystems that are $n$-separable are called fully separable. For example, all graph states of a connected graph and the $n$-qubit GHZ and W states are GME, whereas the maximally mixed state of any number of systems of any dimension is fully separable.

From the definition of $k$-separability, it is clear that all $k$-separable states are also $(k - j)$-separable for all $j \in \{1, \ldots, k - 2\}$, and the sets of states that are at least $k$-separable form a nested structure of convex sets (see, e.g., ref. [60],Ch. 18] for an introduction). In general, the multipartite state space has a rich structure and much progress has been made in understanding it in the past decade. Pertinent developments include the description of high-dimensional multipartite systems using the entropy-vector formalism[61,62], the definition of operational multipartite entanglement measures[63], insights into entanglement transformations via local operations and classical communication[64,65] along with relevant symmetries[66–68], restrictions[69], and potential improvements using quantum metrology[70] for this task. In addition, the recently discovered phenomenon of multi-copy activation of GME[71–74] has added another layer to the problem of unraveling multipartite entanglement structures. In order to better characterize complex quantum states produced in state-of-the-art laboratories in the context of these multi-faceted state-space structures, we hence need suitable tools for the detection of multipartite entanglement.

## GME & $k$-inseparability criteria

The GME and $k$-inseparability criteria that we introduce in this work are associated to an $n$-vertex graph $G$, and are defined as a sum of absolute values of expectation values of the stabilizers corresponding to all vertices of the graph and products of stabilizers for each edge in the graph, i.e.,

$$\mathcal{W}_G^\gamma(\rho) = \sum_{i \in V} |\langle S_i \rangle_\rho| + \gamma \sum_{(i,j) \in E} |\langle S_i S_j \rangle_\rho|, \quad (2)$$

where $\gamma \in [0, 1]$ is a free parameter specifying a different valid GME/$k$-inseparability criterion for each choice, and $\langle A \rangle_\rho : = \text{Tr}(A\rho)$. We will omit the subscript $\rho$ from expectation values whenever the state is clear from context.

The intuition behind this choice of stabilizer subset stems from the use of the *anticommutativity inequality* (Lemma 3 and Proposition 1 in Methods) and the fact that Pauli matrices anticommute, which together lead to the analytic bounds in Eqs. (4) and (8). These properties ensure that whenever two qubits $a$ and $b$ in $\rho$ correspond to adjacent vertices in the underlying graph and belong to different groups of a given $k$-partition of $n$ qubits, any $k$-product state of that $k$-partition satisfies $|\langle S_a \rangle| + |\langle S_b \rangle| + |\langle S_a S_b \rangle| \leq 1$, $|\langle S_a \rangle| + |\langle S_a S_b \rangle| \leq 1$, and $|\langle S_a \rangle| + |\langle S_b \rangle| \leq 1$. In the 5-qubit example shown in Fig. 1, the color-labeled 3-partition of the graph corresponds to evaluating the 3-separability bound for all states of the form $\rho_{145} \otimes \rho_2 \otimes \rho_3$. Such states satisfy, for instance,

$$
\begin{aligned}
&|\langle S_1 \rangle| + |\langle S_2 \rangle| + |\langle S_1 S_2 \rangle| \\
&= |\langle X_1 Z_5 \rangle_{\rho_{145}} \langle Z_2 Z_3 \rangle_{\rho_{23}}| + |\langle Z_1 \rangle \langle X_2 Z_3 \rangle| + |\langle Y_1 Z_5 \rangle \langle Y_2 \rangle| \\
&\leq \sqrt{\langle X_1 Z_5 \rangle^2 + \langle Z_1 \rangle^2 + \langle Y_1 Z_5 \rangle^2} \sqrt{\langle Z_2 Z_3 \rangle^2 + \langle X_2 Z_3 \rangle^2 + \langle Y_2 \rangle^2},
\end{aligned}
\quad (3)
$$

using the Cauchy-Schwarz inequality, with the final expression bounded above by 1 due to the anticommutativity inequality. By contrast, states that are inseparable across this partition can exceed

these bounds since the three sums can reach a maximum of 3, 2, and 2, respectively, e.g., when $\rho = |G\rangle\langle G|$. Because of these structural features arising from our choice of stabilizer subset, our criteria are capable of certifying both GME and the more general $k$-inseparability—something that most conventional stabilizer-based methods cannot achieve[45,46,50].

For any connected graph, our criteria only require measuring $2n - 1 \leq J \leq \frac{n(n+1)}{2}$ (for $\gamma > 0$) or $J = n$ (for $\gamma = 0$) out of the total of $2^n$ ($\leq m$)-body stabilizers of $|G\rangle$ with $m \leq \max_{(i,j)\in E}[d(i) + d(j)]$, and need $\min(n+1, 5) \leq M \leq \frac{n(n+1)}{2}$ (for $\gamma > 0$) or $2 \leq M \leq n$ (for $\gamma = 0$) local measurement settings (choices of $n$-qubit Pauli bases). In particular, for families of $n$-vertex graphs whose maximum degree is independent of $n$ (e.g., chain graphs, regular 1D/2D lattices), our criteria only require measuring $O(1)$-body stabilizers. In contrast, all previous GME/$k$-inseparability witnesses require local measurements on at least $O(n)$ particles simultaneously, except for the witness from Eq. (45) in ref. 46, which cannot certify non-GME $k$-inseparability (see Supplementary Note 1).

The following theorem, which we refer to as the *graph-matching GME criterion*, provides analytic upper bounds of $\mathcal{W}_G^\gamma(\rho)$ for any $k$-separable state $\rho$, such that the violation of any of these bounds detects $k$-inseparability. The proof of the theorem can be found in Methods. We also provide an algorithm that computes the first upper bound in Supplementary Note 4 and the error analysis when applying our GME/$k$-inseparability criteria to experiments in Supplementary Note 10. A graphical illustration of the meaning of the symbols $\overline{G}^{(k)}$, $\overline{E}_{\text{mcm}}^{(k)}$, and $\overline{E}_{\text{match}}^{(k)}$ appearing in Theorem 1 and its proof can be found in Fig. 1, where we also show how the edge set $\overline{E}_{\text{mcm}}^{(k)}$ corresponds to the maximum reduction of the upper bound for $k$-separability. Note that, since the left-hand side of Eq. (4) is a sum of absolute values of stabilizer expectation values, Theorem 1 can be seen as a statement about a collection of linear GME/$k$-inseparability criteria with different combinations of signs for different stabilizer terms.

**Theorem 1.** (Graph-matching GME criterion): Any $n$-qubit ($k \geq 2$)-separable state $\rho$ satisfies

$$\mathcal{W}_G^\gamma(\rho) \leq n + \gamma|E| - R_k^\gamma \leq n + \gamma(|E| - k + 1) - 1, \quad (4)$$

for all $\gamma \in [0, 1]$, where

$$R_k^\gamma := \min_{\text{all } k\text{-cuts}} \left( \gamma|\overline{V}^{(k)}| + (1 - \gamma)|\overline{E}_{\text{mcm}}^{(k)}| \right), \quad (5)$$

and $\overline{V}^{(k)}$ is the vertex set of the subgraph $\overline{G}^{(k)}$ of which the edge set $\overline{E}^{(k)}$ corresponds to the edges that a $k$-cut of the full graph $G$ removes, and $\overline{E}_{\text{mcm}}^{(k)}$ denotes the maximum cardinality matching of $\overline{G}^{(k)}$.

Theorem 1 implies that if $\mathcal{W}_G^\gamma(\rho) > n + \gamma|E| - R_k^\gamma$ or $n + \gamma(|E| - k + 1) - 1$ for any $\gamma \in [0, 1]$ (and if $k = 2$), then $\rho$ is $k$-inseparable/not $k$-separable (is GME). Hence, the corresponding optimal GME/$k$-inseparability criterion is given by

$$\max_{0 \leq \gamma \leq 1} \mathcal{W}_G^\gamma(\rho) - \gamma|E| + R_k^\gamma - n > 0. \quad (6)$$

In general, the optimal choice of $\gamma$ for achieving the maximum in Eq. (6) depends on two factors: (i) the measured values of the two summation terms in $\mathcal{W}_G^\gamma(\rho)$, which vary across different experiments, and (ii) the number of edges $|E|$ and the reduction term $R_k^\gamma$, which behave differently for different underlying graphs. Despite the criterion in Eq. (6) having a seemingly linear form, the term $R_k^\gamma$ is in general nonlinear in $\gamma$, and while the optimal choice of $\gamma$ is $\gamma = 0$ or $\gamma = 1$ for some states (e.g., 2D cluster states), this is not always the case.

**Observation 1.** There exist states $\rho$ and graphs $G$ such that the optimal GME/$k$-inseparability criterion is achieved for $\gamma \in (0, 1)$.

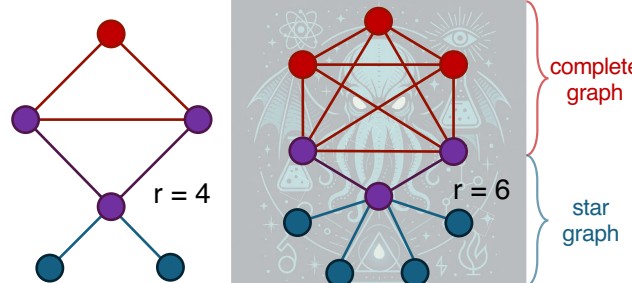

**Fig. 2 | Cthulhu graphs: The parameter $r$ represents both the number of vertices in the "head" subgraph (red and purple vertices) and the degree of the central vertex in the star subgraph containing the "tentacles" (blue and purple vertices).** These graphs are defined such that the "head" subgraph contains an $(r - 1)$-vertex complete graph with two adjacent vertices being the leaves of the star subgraph. Their noisy graph states are examples where the optimal $r$-inseparability criteria is achieved for $\gamma \in (0, 1)$.

The examples for which we made this observation are mixed states $\rho = \frac{p}{2^n}\mathbb{1} + (1 - p)|G\rangle\langle G|$ with $0 \leq p \leq 1$ obtained by adding white noise to particular graph states $|G\rangle$ whose underlying graphs are what we call *Cthulhu graphs*. Such graphs, parametrized by an integer $r \geq 3$, consist of an $(r - 1)$-vertex complete graph (the "head") attached to a degree-$r$ tree graph (the "tentacles"), as illustrated in Fig. 2. In Supplementary Note 3, we show that for $r = 4$ and $r \geq 6$, the optimal choice of $\gamma$ in Eq. (6) to detect the state $\rho$ as being $r$-inseparable is $\gamma = (\lfloor\frac{r}{2}\rfloor - 1)/\lfloor\frac{r}{2}\rfloor$ if $p$ lies in the range

$$\frac{2\lceil\frac{r}{2}\rceil}{r(r-1)+2} < p < \frac{2[(r+1)\lfloor\frac{r}{2}\rfloor - r]}{(r^2 + 3r - 2)\lfloor\frac{r}{2}\rfloor - r^2 + r - 2}. \quad (7)$$

This intermediate value of $\gamma$ naturally arises because Cthulhu graphs merge components whose optimal $\gamma$ lie at the extremal points 0 and 1 (see Supplementary Note 7), causing the optimal choice for the combined structure to interpolate between them.

For state-diagnostic purposes, it may be sufficient to assess a given state's (in)separability with respect to a fixed $k$-partition (in contrast to $k$-(in)separability, which considers all $k$-partitions). For this case, we also provide a family of criteria to determine such (in)separability in Lemma 1, whose proof follows from that of Theorem 1 (see Methods).

**Lemma 1.** (Fixed $k$-partition inseparability criterion): Any $n$-qubit state $\rho$ that is separable with respect to a specific ($k \geq 2$)-partition satisfies

$$\mathcal{W}_G^\gamma(\rho) \leq n + \gamma\left(|E| - |\overline{V}^{(k)}|\right) - (1 - \gamma)|\overline{E}_{\text{mcm}}^{(k)}|, \quad (8)$$

for all $\gamma \in [0, 1]$, with $\overline{V}^{(k)}, \overline{E}_{\text{mcm}}^{(k)}$ defined in Theorem 1.

Apart from noise and decoherence, graph states prepared in experiments may also differ from those described above in terms of the local bases with respect to which they are defined in section "Background and notation". Since local unitary (LU) transformations cannot change entanglement of any state and the proof of Theorem 1 is unaffected by LU conjugations of all of the stabilizers $S_i$ and $S_iS_j$ in Eq. (2), GME/$k$-inseparability of such states can also be efficiently detected by our criteria by adapting the stabilizers by LU conjugation. Similarly, we can target stabilizer states, a larger family of states that includes the set of graph states as a subset. All stabilizer states are equivalent to graph states up to local Clifford (LC) operations—the subset of local unitaries that map the Pauli group to itself[56,57]. For example, the $n$-qubit GHZ state can be obtained from the graph state for the star graph by application of Hadamard gates $H$, i.e., $|\text{GHZ}_n\rangle = \mathbb{1} \otimes H^{\otimes n-1}|G\rangle$, where

the first qubit corresponds to the central vertex of the star graph $G$. Let us summarize the above in the following remark.

**Remark 1.** One can minimize $k$ of the certified $k$-inseparability of a state by optimizing over LU conjugations of the stabilizers $S_i$ and $S_iS_j$ in Eq. (2). This maximizes the amount of information about multipartite entanglement that can be gained with our GME/$k$-inseparability criteria in any state, especially for states close to stabilizer states.

In addition, LC operations generate equivalence classes of graph states: two graph states $|G\rangle$ and $|G'\rangle$ are *LC-equivalent* if they differ only by a sequence of LC operations, with their graphs related by *local complementations*[57]. This freedom can be used to optimize our criteria under restricted measurements. One may choose an LC-equivalent representative $|G'\rangle = \otimes_{a=1}^{n} C_a |G\rangle$, where $C_a$ is a Clifford unitary, whose graph has the smallest maximum degree in the class, thereby reducing the stabilizer weight in Eq. (2). The witness is then evaluated on $G' = (V', E')$, replacing $S_i$ for $i \in V'$ and $S_iS_j$ for $(i,j) \in E'$ by the LC-conjugated stabilizers of $|G'\rangle$, $(\otimes_{a=1}^{n} C_a^\dagger) S_i (\otimes_{a=1}^{n} C_a)$ and $(\otimes_{a=1}^{n} C_a^\dagger) S_iS_j (\otimes_{a=1}^{n} C_a)$, which stabilize $|G\rangle$ but are no longer the vertex generators or edge-generator products of the original graph. In Supplementary Note 5, we provide an example of how applying local complementations (and the corresponding LC operations) to obtain a graph with a lower maximum degree reduces the maximum stabilizer weight required by our criteria. Alternatively, to minimize the total number of stabilizer terms that need to be measured, one may select a *minimum-edge representative* (MER) within the LC-equivalence class, which minimizes the number of edge terms in Eq. (2). Methods for finding MERs can be found in ref. 75.

**Remark 2.** In some cases, applying local complementations to the underlying graph can reduce the number or maximum weight of stabilizers required by our criteria.

Although graph states serve as targets for the construction of the criteria, and although the latter work particularly well for (noisy) graph states (as we shall demonstrate numerically for realistic experimental situations in section "Experimental proposal and simulations" and analytically for white-noise-added graph states in Supplementary Note 7) and stabilizer states (up to LC), our GME/$k$-inseparability criteria can also certify multipartite entanglement for other states. For example, we can certify GME in non-stabilizer states that are LU equivalent to (noisy) Dicke states (see Supplementary Note 7), which leads us to the following remark.

**Remark 3.** While our criteria are defined with respect to a specific graph $G$, for which the state $|G\rangle$ achieves the maximum value $n + \gamma|E|$ of $\mathcal{W}_G^\gamma$, it remains a valid entanglement criterion for any $n$-qubit state (including non-stabilizer states): if Eq. (4) is violated, GME/$k$-inseparability is certified, regardless of the underlying state.

At the same time one should note that for certain $n$-qubit states, such as GHZ states (which are LC-equivalent to both star- and complete-graph states), any method that can certify their GME must measure at least one $n$-body observable[76]. Thus, it is clear that no criteria using only constant-weight observables (including those presented here) can detect GME in all $n$-qubit states.

Note that the second bound in Eq. (4), which is looser than the first bound for certain graph states, recovers the witness in Eq. (45) of ref. 46 if we set $\gamma = 0$, in which case no $k$-inseparability can be detected for $k > 2$ that is not GME. While the first upper bound is generally tighter, the computational cost of calculating $R_k^\gamma$ using the (potentially suboptimal) algorithm presented in Supplementary Note 4 grows exponentially with $n$, as enumerating all $k$-partitions of $n$ vertices takes $O(k^n)$. This overhead does not arise for the fixed $k$-partition inseparability criteria (Lemma 1). In addition, for each partition, the maximum-cardinality matching is determined with a cost of $O(|\overline{V}^{(k)}|^2 \cdot |\overline{E}^{(k)}|)$ using the most widely used algorithm[77] or $O(|\overline{V}^{(k)}|^{1/2} \cdot |\overline{E}^{(k)}|)$ using the

most efficient algorithm known to date[78]. Hence, for large $n$, the second inequality in Eq. (4) can serve as a heuristic $k$-separability criterion that is easy to verify.

As mentioned at the beginning of this section, our criteria generally only require measuring $2n - 1 \le J \le \frac{n(n+1)}{2}$ of the ($\le m$)-body stabilizers of the graph state $|G\rangle$ with $m \le \max_{(i,j) \in E}[d(i) + d(j)]$, and generally need $\min(n+1, 5) \le M \le \frac{n(n+1)}{2}$ local measurement settings. In the next section, we will show that using SDP, one can potentially further bring down the number and maximum weight of the measured stabilizers to as low as $J = n$ and $m \le \max_{i \in V} d(i) + 1$, using as few as $2 \le M \le n$ local measurement settings.

## SDP for incomplete measurements

In some experimental situations, even more stringent measurement constraints might apply. For example, it may occur that only the stabilizer generators $S_i$ are accessible, but not the stabilizer products $S_iS_j$, meaning that not all terms of our GME/$k$-inseparability criteria in Eq. (2) are measurable (see section "Experimental proposal and simulations"). One can of course lower bound these terms trivially by zero. However, this will most likely lead to not certifying any $k$-inseparability for $k < n$ since $\mathcal{W}_G^\gamma(\rho)$ would be significantly underestimated. Given access to expectation values of up to $m$-body correlators, we can potentially obtain non-trivial lower bounds for these stabilizer terms via an SDP with constraints based on the measurable correlators and on $\rho$ being a density matrix. The simplest approach is to linearize the following optimization problem:

$$\min_\rho |\text{Tr}(S_{v_1}S_{v_2}\rho)| \tag{9a}$$

$$\text{subject to } |\text{Tr}(S_{v_j}\rho) - b_j| \le \varepsilon_j \text{ for } j \in \{1, 2\}, \tag{9b}$$

$$\text{Tr}(\rho) = 1, \rho \ge 0, \tag{9c}$$

and solve the corresponding dual SDP problem, with $b_j$ being the measured expectation values of $S_{v_j}$ and $\varepsilon_j$ their statistical uncertainty. More generally, one can incorporate more experimentally inaccessible terms from Eq. (2) into the objective function, and more measurable observables into the constraints to get potentially tighter lower bounds for the sum of those inaccessible terms. The general form of such optimizations and their associated SDPs can be found in Methods.

The advantage of applying SDP here is that the numerically obtained dual optimal solutions are always faithful lower bounds of those experimentally inaccessible terms in our criteria due to *weak duality*. In addition, by proving *strong duality* holds for our general SDP problem in Eqs. (23a–d), we are promised to get numerically tight lower bounds (see Methods for more details).

In section "Certifying GME/k-inseparability in simulated states", we solve the dual SDP that corresponds to Eqs. (9a–c) to lower bound each term $|\langle S_iS_j\rangle|$ that enters our GME/$k$-inseparability criteria with only the measured expectation values $\langle S_i\rangle$ and $\langle S_j\rangle$ as constraints. Although the certified entanglement is generally lower than when all $\langle S_iS_j\rangle$ are measured, the certified GME/$k$-inseparability is often comparable (see, e.g., Supplementary Table S4). Thus, incorporating this SDP technique allows us to certify GME/$k$-inseparability even under more restrictive measurement conditions.

Regarding scalability, the SDP can be solved efficiently on a standard laptop using the MOSEK solver in MATLAB for reduced states of up to 12 qubits. Therefore, this method readily applies to all graph states satisfying $\max_{(i,j) \in E}[d(i) + d(j)] \le 12$, including many graph states of practical importance in quantum information science, such as all 1D to 3D cluster states[20], all ring-graph states[42], and many tree-graph

states[43,44]. Beyond 12 qubits, while solving the SDP may become intractable with conventional interior-point methods, alternative methods with better scalability[79], such as augmented Lagrangian methods, can be employed for larger problems, although the development of more stable software implementations is still required.

In the next section, we will justify the measurement restrictions—limited to at most $O(1)$-body observables—that we have been considering so far with a concrete experimental scenario. Furthermore, we will show that our GME and $k$-inseparability criteria perform well even for graph states simulated under realistic experimental conditions.

## Experimental proposal and simulations

To showcase possible applications of the presented GME and $k$-inseparability criteria, we propose to evaluate the criteria on graph states consisting of microwave photonic qubits. Recent experiments with superconducting circuits have demonstrated the capability to generate large-scale graph states comprising tens of photonic qubits[36–39]. However, characterizing the quality of the generated states is still challenging, as high-fidelity single-photon detectors in the microwave regime are still the subject of ongoing research[80–82]. In state-of-the-art experiments, microwave photons are first amplified using near-quantum-limited amplifiers and then detected via heterodyne measurements. Due to vacuum and thermal noise added during the amplification process, the signal-to-noise ratio (SNR) of single-photon measurements in the microwave regime is typically limited to around $\eta \approx 0.2 - 0.4$[38,39]. Due to the exponential scaling of the SNR with the weight of the Pauli observable (see Supplementary Note 8), the measured Pauli observables are limited to weight 5[39]. This makes the criteria proposed in this work particularly attractive, as only low-weight Pauli expectation values are required.

Building on early proposals for sequential generation of graph states[83] and recent demonstrations of cluster-state generation using superconducting circuits[38], we consider a physical system comprising of multiple superconducting transmon qubits, each tunably coupled to a waveguide, as illustrated in Fig. 3 (see also Supplementary Note 9 for a more detailed introduction). By combining single- and two-qubit gates on the transmons with controlled emission, such a system can generate various graph states, including one- and two-dimensional cluster states, ring-graph states, and tree-graph states, whose quality can be benchmarked using the entanglement criteria presented in this work.

## Certifying GME/$k$-inseparability in simulated states

Following the numerical approach detailed in ref. 38 and Supplementary Note 9, we simulate the generation protocol for cluster, ring-graph, and tree-graph states of varying sizes. The computed $k$-inseparability of the simulated states are summarized in Table 1 and Supplementary Tables S3–S5 in Supplementary Note 11. Specifically, we compare using our GME/$k$-inseparability criteria—when all terms in Eq. (4) are measured—with that obtained using the witness from Eq. (45) of ref. 46 [hereafter referred to as the "TG45 witness"], which is, to the best of our knowledge, the only witness/criterion in the literature that requires measurements of at most $O(1)$-body observables. In the same table, we also show the certified GME/$k$-inseparability using our criteria when only the stabilizer generators [terms in the first sum in Eq. (4)] are measured and the terms $|\langle S_i S_j \rangle|$ are unmeasured but lower bounded by the dual SDP in section "SDP for incomplete measurements" using only the expectation values $\langle S_i \rangle$ and $\langle S_j \rangle$ as constraints. For completeness, we also include the GME/$k$-inseparability witnesses of ref. 47, even though this method generally requires to measure at least $O(2^{n/c})$ stabilizers with a maximum degree of up to $O(n)$, where $c$ denotes the chromatic number of the underlying graph.

For all of the simulated graph states, whenever the TG45 witness detects GME, our criterion always detects GME. However, there are many states for which the TG45 witness cannot certify GME (and is

therefore inconclusive), whereas our criteria can still certify $k$-inseparability for a relatively small $k$, suggesting that a significant amount of multipartite entanglement is still present. This observation is particularly prominent for the simulated ring-graph states (see Table 1) where our criterion detects GME in the 7- and 8-qubit states, while the TG45 witness cannot. Regarding the witnesses of ref. 47, they perform slightly better than our criteria, but their required measurements are more experimentally demanding and become infeasible in the experimental platforms we consider for large $n$.

Furthermore, even without measuring the $|\langle S_i S_j \rangle|$ terms, our GME/$k$-inseparability criteria—where the second sum in Eq. (4) bounded by SDP—can still certify $k$-inseparability at a level comparable to that of our full criteria (with all terms measured), thus achieving similar certification power while using much fewer measured stabilizers. This also highlights the advantage of our criteria: they can already outperform the TG45 witness using just the measurement data associated with the stabilizer generators of the underlying graph state.

The fidelities $F$ between the simulated states and the corresponding ideal, maximally entangled, graph states are also shown in Table 1 and Supplementary Tables S3–S5 in Supplementary Note 11. Since $F > 0.5$ for all simulated states, all simulated states are GME[45,49]. However, since our GME and $k$-inseparability criteria only use at most $(2n - 1)/2^n$ of the total stabilizers, we expect to not be able to certify GME for states with fidelities not close to 1. Also, we note that we obtain the fidelities using full density matrices from our numerical simulations. However, under the realistic experimental restrictions that we consider in this work, the fidelity is not an accessible quantity since estimating/lower bounding it requires measuring up to $O(n)$-body observables (see Supplementary Note 1).

## Noise sensitivity of the GME/$k$-inseparability criteria

To study how the certified entanglement depends on noise parameters, we further simulate the generation of a $5 \times 2$ cluster state with varying noise parameters. The two common error sources in experiments are decoherence errors and leakage errors. For decoherence errors, we simultaneously scale the coherence times of both source modes. The parameter we choose to quantify the decoherence error is $\tau_{\text{emit}}/\tau_{\text{coh}}$, where $\tau_{\text{emit}}$ is the time taken to emit a pair of photons, and $\tau_{\text{coh}} := \min(T_1^{So_1,g-e}, T_2^{So_1,g-e}, T_1^{So_2,g-e}, T_2^{So_2,g-e})$ is the smallest coherence time. In ref. 38, $\tau_{\text{emit}} = 650$ ns and $\tau_{\text{coh}} = 22$ μs. The leakage errors lead to residual population in the second excited state of the transmon after a two-qubit CZ gate or a controlled-emission CNOT gate. We simultaneously vary the leakage error values of the CZ and CNOT gates around the experimental values $L_{CZ} = 2\%$ and $L_{\text{CNOT}} = 1\%$, as reported in ref. 38. The full list of parameters used in the simulation is given in Supplementary Note 9.

The certified $k$-inseparability qualitatively follows the corresponding state infidelities, see Fig. 4. Both increasing decoherence errors and leakage errors increase $k$, indicating a reduction in the amount of certifiable entanglement. This demonstrates that our criterion is a useful diagnostic tool for the performance of the experiment, especially as the state fidelity is difficult to obtain experimentally. It is also worth noting that the certified $k$ value is always odd, for $k > 2$, meaning the states would always be separated into an even number of parties. This is possibly due to there being a pair of source transmons, but confirming this requires further investigation.

## Discussion

We have presented a new set of criteria for detecting GME/$k$-inseparability and fixed $k$-partition inseparability that can be used in measurement-restricted settings where other existing criteria cannot be evaluated. Our criteria are defined by graphs and the associated graph states, and are ideally suited for characterizing multipartite entanglement in the latter, or in more general stabilizer states with few

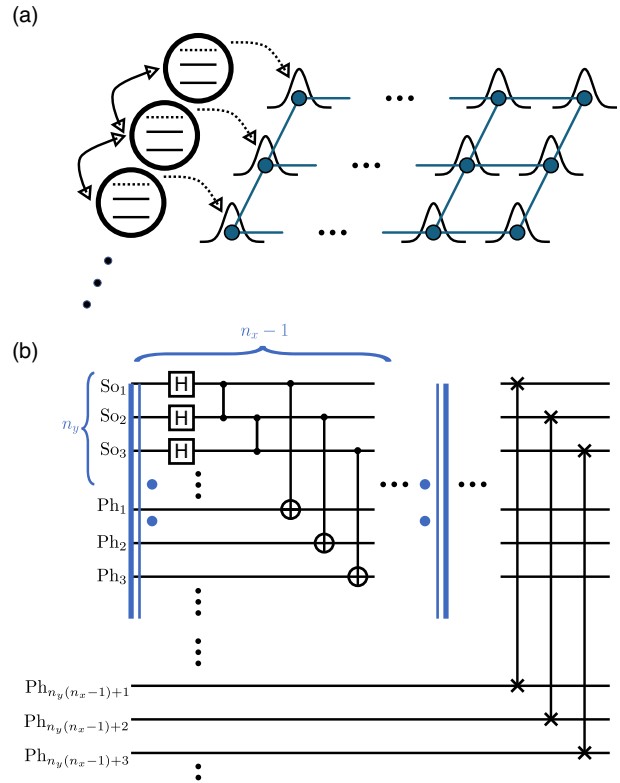

**Fig. 3 | Schematic and quantum circuit for cluster state generation. a** Schematic of sequentially generated microwave photonic cluster state in two dimensions. The setup consists of a linear array of tunably interacting transmon qutrits. Arbitrary single- and two-qubit gates on the lowest two levels and the third level can be used for controlled emission of microwave photons. **b** Quantum circuit for creating $n_x \times n_y$ two-dimensional cluster state. In the circuit, $So_i$ denotes the $i$'th source transmon, whereas $Ph_i$ denotes the $i$'th generated photonic qubit. The double blue lines and two dots indicate the block of circuit to be repeated.

simple measurements. Nevertheless, our method can also be employed independently of the underlying state.

To demonstrate the flexibility and performance of our approach, we derived analytical white-noise thresholds for detecting $k$-inseparability of various noisy graph states, as well as noisy non-stabilizer states that are LU-equivalent to Dicke states. We further conducted a series of numerical experiments using realistic parameters for setups generating time-bin entangled microwave photonic qubits. The results indicate that our method can reliably detect GME and thus help characterize devices used for generating complex quantum states that serve as resources for, e.g., quantum computation, quantum sensing, and quantum networks. In all of these applications, multipartite entanglement is of central importance, and its certification can serve as a benchmark that provides at least partial assurance of device functionality and quality of control when full state or device characterization is impractical.

While promising, our methods have yet to be tested under real laboratory conditions, which typically come with additional technological challenges. Some of these difficulties, such as further restrictions on the number or type of measurable observables, may be ameliorated by employing SDP techniques, as discussed in the manuscript, while other complications may motivate further refinements in the design of entanglement-detection criteria. We therefore envisage tests on actual hardware as a logical next step and an opportunity for further research.

Beyond these practical considerations, our GME/$k$-inseparability criteria admit a natural interpretation as a Hamiltonian $H_G = \sum_{i \in V} S_i + \gamma \sum_{(i,j) \in E} S_i S_j$, whose structure closely resembles families of

local Pauli Hamiltonians studied in the many-body physics literature, including those appearing in the context of symmetry-protected topological (SPT) phases[84,85]. In the "pivot Hamiltonian" framework[85], Hamiltonians of the 1D cluster model can be obtained by conjugating a classical Ising model $H_{\text{Ising}} = \sum_{i \in V} X_i + \gamma \sum_{(i,j) \in E} X_i X_j$ with a unitary generated by a graph-local Ising-type pivot, suggesting a structural analogy with the Hamiltonian form considered here. This procedure can generate Hamiltonians whose ground states exhibit nontrivial SPT order[85]. Our entanglement criteria establish that any state−pure or mixed−whose energy violates

$$|\langle H_G \rangle_\rho| \le \mathcal{W}_G^\gamma(\rho) \le n + \gamma|E| - R_k^\gamma, \tag{10}$$

is necessarily GME/$k$-inseparable. This complements the conventional perspective in many-body physics, which focuses predominantly on ground states or low-energy excited states. The Hamiltonian structure underlying our witness is particularly noteworthy, as it motivates a broader framework for constructing entanglement witnesses from the many-body physics perspective. This connection suggests that such entanglement-witnessing Hamiltonians may be systematically derived by combining well-understood many-body constructions with entanglement theory, opening new avenues for exploring the interplay between multipartite entanglement and quantum many-body phenomena.

## Methods
### Proof of Theorem 1 and Lemma 1
To prove Theorem 1 and Lemma 1, we need to first state the following lemma and propositions. The proof of Lemma 3 can be found in ref. 86 (Theorem 1) while Propositions 1 and 2 are proven in Supplementary Note 2.

**Lemma 3.** (Anticommutativity bound[86]): Let $\{E_i\}_{i=1}^{d^2}$ be an orthonormal self-adjoint basis of $d \times d$ complex matrices (i.e., $\text{Tr}(E_i E_j) = d\delta_{ij}$) and let $\Omega \subseteq \{1, ..., d^2\}$ such that $\frac{1}{2}\sqrt{\sum_{i \ne j \in \Omega} \langle \{E_i, E_j\} \rangle_\rho^2} \le \mathcal{K}$ where $\{A, B\} := AB + BA$. Then,

$$\sum_{i \in \Omega} \langle E_i \rangle_\rho^2 \le \max_{i \in \Omega} \langle E_i^2 \rangle_\rho + \mathcal{K} \tag{11}$$

for any state $\rho \in \mathcal{D}(\mathbb{C}^d)$.

**Proposition 1.** Let $\{A_i\}_{i=1}^m$ and $\{B_i\}_{i=1}^m$ be subsets of orthonormal self-adjoint bases of $d_1 \times d_1$ and $d_2 \times d_2$ complex matrices, respectively, such that $\{A_i, A_j\} = 2\delta_{ij}\mathbb{1}_{d_1}$ and $\{B_i, B_j\} = 2\delta_{ij}\mathbb{1}_{d_2}$ for all $i, j$. Then, the expectation values of $A_i$ and $B_i$ with respect to any quantum states $\rho \in \mathcal{D}(\mathbb{C}^{d_1})$ and $\sigma \in \mathcal{D}(\mathbb{C}^{d_2})$ must satisfy $\sum_{i=1}^m |\langle A_i \rangle_\rho \langle B_i \rangle_\sigma| \le 1$.

**Proposition 2.** Any $k$-cut of a connected graph must remove at least $k - 1$ edges that are shared among at least $k$ vertices.

**Proof.** (Proof of Theorem 1 & Lemma 1): Since $\mathcal{W}_G^\gamma(\rho)$ is a convex function of $\rho$, we only need to prove that Eq. (4) is satisfied by all $k$-separable pure states $\rho = |\psi\rangle\langle\psi| = \bigotimes_{j=1}^k |\psi^{(j)}\rangle\langle\psi^{(j)}|$. We consider a fixed partition of $n$ qubits into $k \ge 2$ groups such that the $j$-th group of qubits, denoted by the labeling set $P_j$, corresponds to the $j$-th tensor factor of $|\psi\rangle = \bigotimes_{j=1}^k |\psi^{(j)}\rangle$. By matching the labels of the $n$ qubits and the $n$ vertices of the graph $G$, we consider the same partition to the graph $G$ such that $\bigcup_{j=1}^k P_j = V$ where $k \ge 2$ and $P_i \cap P_j = \varnothing \ \forall i \ne j$.

Let us define $\overline{E}^{(k)} := \{(a, b) \in E \mid a \in P_i, b \in P_j \text{ and } i < j\}$. In graph theory language, $\overline{E}^{(k)}$ contains the edges removed by a $k$-cut of the full graph $G$. The set of vertices involved in the edges in $\overline{E}^{(k)}$ is $\overline{V}^{(k)}$. They together define the subgraph $\overline{G}^{(k)} = (\overline{V}^{(k)}, \overline{E}^{(k)})$ of $G$.

We first prove the bound in Eq. (4) holds for $\gamma = 1$. Let $\overline{E}_{\text{match}}^{(k)} \subset \overline{E}^{(k)}$ be a maximal matching of $\overline{G}^{(k)}$. For each edge $(a, b) \in \overline{E}_{\text{match}}^{(k)}$, it must

hold that vertices/qubits $a \in P_i$ and $b \in P_{j \neq i}$ belong to different groups in the $k$-partition. Since $|\psi\rangle = \otimes_{j=1}^{k} |\psi^{(j)}\rangle$ is separable across $P_i$ and $P_{j \neq i}$, we can use Proposition 1 to show that the corresponding stabilizers of the graph state $|G\rangle$ satisfy

$$
|\langle S_a \rangle| + |\langle S_b \rangle| + |\langle S_a S_b \rangle|
$$
$$
= |\langle X_a Z_{N(a) \cap P_i} \rangle \langle Z_b Z_{N(a) \cap P_{\neg i} \setminus \{b\}} \rangle| \tag{12a}
$$
$$
+ |\langle Z_a Z_{N(b) \cap P_i \setminus \{a\}} \rangle \langle X_b Z_{N(b) \cap P_{\neg i}} \rangle|
$$
$$
+ |\langle Y_a Z_{N(a) \Delta N(b) \cap P_i \setminus \{a\}} \rangle \langle Y_b Z_{N(a) \Delta N(b) \cap P_{\neg i} \setminus \{b\}} \rangle| \tag{12b}
$$
$$
\leq 1,
$$

**Table 1 | Comparison of certified multipartite entanglement in simulated $n$-qubit ring-graph states using different witnesses/criteria**

| $n$ | 46's witness | 47's witness | Our criteria | Our criteria (SDP) | Fidelity |
|---|---|---|---|---|---|
| 4 | GME | GME | GME | GME | 0.838 |
| 5 | GME | GME | GME | GME | 0.810 |
| 6 | GME | GME | GME | GME | 0.785 |
| 7 | / | GME | GME | 3-insep | 0.745 |
| 8 | / | GME | GME | 3-insep | 0.719 |
| 9 | / | 3-insep | 3-insep | 3-insep | 0.683 |
| 10 | / | 3-insep | 3-insep | 3-insep | 0.659 |
| 11 | / | 3-insep | 3-insep | 3-insep | 0.626 |
| 12 | / | 3-insep | 4-insep | 5-insep | 0.607 |

The first column shows the GME certification results using the witness from Eq. (45) in ref. 46. The second column shows certification results using the witness in Eq. (15) of ref. 47, optimized over all 3-colorings of odd-$n$ ring graphs. These witnesses require at least $O(2^{n/3})$ stabilizers, with a maximum weight up to $O(n)$. The third column reports GME/$k$-inseparability certified by our criteria with all terms in Eq. (2) measured. The fourth column shows results from our criteria with only the stabilizer generators measured, and all $|\langle S_i S_j \rangle|$ in Eq. (2) lower bounded by the dual SDP in section "SDP for incomplete measurements". The last column gives the fidelities with the ideal graph states, calculated from the full simulated density matrices.

where $Z_A := \prod_{i \in A} Z_i$ for $A \subseteq V$, $N(a)$ denotes the neighborhood of vertex $a$, $\Delta$ denotes the symmetric difference, and $P_{\neg i} := V \setminus P_i$.

We then consider the remaining vertices and edges in $\overline{G}^{(k)}$ that are not in the maximal matching $\overline{E}_{\text{match}}^{(k)}$. Let us define $\overline{V}_{\text{match}}^{(k)}$ to be the set of vertices in $\overline{E}_{\text{match}}^{(k)}$. Since every remaining edge in $\overline{E}^{(k)} \setminus \overline{E}_{\text{match}}^{(k)}$ must connect to one of the vertices in $\overline{E}_{\text{match}}^{(k)}$ by the definition of a maximal matching, every remaining vertex $a \in \overline{V}^{(k)} \setminus \overline{V}_{\text{match}}^{(k)}$ must have at least one edge that connects itself to a vertex $b \in \overline{V}_{\text{match}}^{(k)}$. Also, since every edge $(a,b) \in \overline{E}^{(k)}$ corresponds to a cut in the $k$-partition, it must hold that vertices/qubits $a \in P_i$ and $b \in P_{j \neq i}$ belong to different groups in the partition. As the stabilizer term $|\langle S_b \rangle|$ of $b \in \overline{V}_{\text{match}}^{(k)}$ is already paired up with other stabilizer terms in Eq. (12a), the remaining unpaired stabilizer terms corresponding to the edge connecting $a \in P_i \cap (\overline{V}^{(k)} \setminus \overline{V}_{\text{match}}^{(k)})$ and $b \in P_{j \neq i} \cap \overline{V}_{\text{match}}^{(k)}$ satisfy

$$
|\langle S_a \rangle| + |\langle S_a S_b \rangle| = |\langle X_a Z_{N(a) \cap P_i} \rangle \langle Z_b Z_{N(a) \cap P_{\neg i} \setminus \{b\}} \rangle|
$$
$$
+ |\langle Y_a Z_{N(a) \Delta N(b) \cap P_i \setminus \{a\}} \rangle \langle Y_b Z_{N(a) \Delta N(b) \cap P_{\neg i} \setminus \{b\}} \rangle| \leq 1, \tag{13}
$$

where we use the tensor product structure of $|\psi\rangle$ and Proposition 1 again.

We can now show that the first inequality in Eq. (4) holds for $\gamma = 1$ by first noticing that $|\langle P \rangle| \leq 1$ for all $n$-qubit Pauli operators $P \in \mathcal{P}_n$. Therefore, the maximum value of $\mathcal{W}_G^{\gamma=1}(\rho)$ in Eq. (2) is $n + |E|$. For $k$-separable $\rho = \otimes_{j=1}^{k} |\psi^{(j)}\rangle \langle \psi^{(j)}|$, we have shown that the sum of some terms in Eq. (2)—in a group of 2 or 3 terms—can be bounded by 1. Each of these bounded sums reduces the upper bound of $\mathcal{W}_G^{\gamma=1}(\rho)$ from $n + |E|$ by 1 or 2. Specifically, the sum of stabilizer terms corresponding to each edge in $\overline{E}_{\text{match}}^{(k)}$ [see Eq. (12b)] gives a reduction of 2, whereas the stabilizer sum corresponding to each vertex in $\overline{V}^{(k)} \setminus \overline{V}_{\text{match}}^{(k)}$ together with one of its connecting edges [see Eq. (13)] reduces the upper bound by 1. Therefore, the total reduction in the upper bound of $\mathcal{W}_G^{\gamma=1}(\rho)$ is given by

$$
R_k^{\gamma=1} = 2 |\overline{E}_{\text{match}}^{(k)}| + |\overline{V}^{(k)} \setminus \overline{V}_{\text{match}}^{(k)}|
$$
$$
= |\overline{V}_{\text{match}}^{(k)}| + (|\overline{V}^{(k)}| - |\overline{V}_{\text{match}}^{(k)}|) = |\overline{V}^{(k)}|, \tag{14}
$$

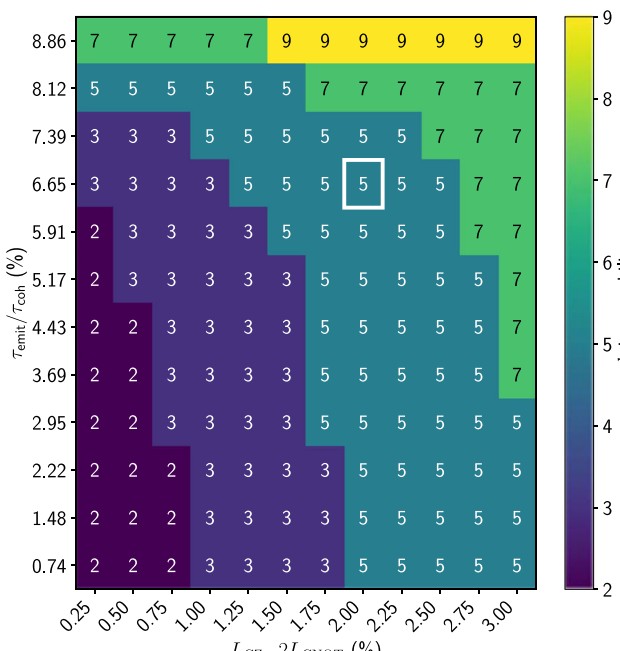
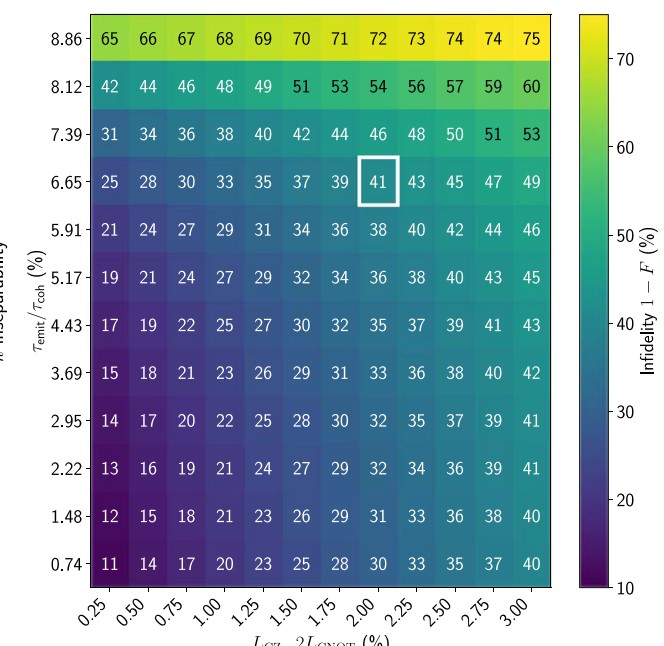

**Fig. 4 | Certified $k$-inseparability and infidelity (to the ideal cluster state) of the simulated 5 × 2-qubit 2D cluster state for different noise parameters.** In the x-axis, leakage errors $L_{\text{CZ}}$ and $L_{\text{CNOT}}$, which are the dominant coherent errors, are varied. In the y-axis, the coherence times of the source transmons are varied. The parameter $\tau_{\text{emit}}/\tau_{\text{coh}}$ represents the photon emission versus coherence times ratio (see text for definition). The points corresponding to the experimental noise parameters from ref. 38 are marked with white squares.

where we used the fact that $|\overline{V}_{match}^{(k)}| = 2|\overline{E}_{match}^{(k)}|$ since any matching consists of pairwise non-adjacent edges. Thus, for all states of this tensor product structure $\rho = \otimes_{j=1}^{k} |\psi_i^{(j)}\rangle\langle\psi_i^{(j)}|$ which corresponds to a particular $k$-partition of the graph $G$, it holds that

$$\mathcal{W}_G^{\gamma=1}(\rho) \leq n + |E| - R_k^{\gamma=1} = n + |E| - |\overline{V}^{(k)}|. \tag{15}$$

Let us now move on to prove the bound in Eq. (4) holds for $\gamma = 0$. We will upper bound $\mathcal{W}_G^{\gamma=0}(\rho) = \sum_{i\in V}|\langle S_i\rangle|$ by considering again the maximal matching of $\overline{G}^{(k)}$. For every edge $(a,b) \in \overline{E}_{match}^{(k)}$, it holds that vertices/qubits $a \in P_i$ and $b \in P_{j\neq i}$. We again consider pure states $\rho = \otimes_{j=1}^{k} |\psi^{(j)}\rangle\langle\psi^{(j)}|$ with the tensor product structure following the same $k$-partition as before. By Proposition 1,

$$\begin{aligned}|\langle S_a\rangle| + |\langle S_b\rangle| &= |\langle X_a Z_{N(a)\cap P_i}\rangle\langle Z_b Z_{N(a)\cap P_{-i}\setminus\{b\}}\rangle| \\ &+ |\langle Z_a Z_{N(b)\cap P_i\setminus\{a\}}\rangle\langle X_b Z_{N(b)\cap P_{-i}}\rangle| \leq 1.\end{aligned} \tag{16}$$

Since $|\langle S_i\rangle| \leq 1 \; \forall \; i$, the maximum value $\mathcal{W}_G^{\gamma=0}(\rho)$ can take is $n$, while each pair of sums in Eq. (16) corresponding to an edge in $\overline{E}_{match}^{(k)}$ reduces the upper bound by 1. The latter contributes to a reduction of $R_k^{\gamma=0} = |\overline{E}_{match}^{(k)}|$ in total. To maximize the reduction, we can choose the maximum-cardinality matching $\overline{E}_{mcm}^{(k)}$ as $\overline{E}_{match}^{(k)}$. Therefore, for all states of this tensor product structure $\rho = \otimes_{j=1}^{k} |\psi_i^{(j)}\rangle\langle\psi_i^{(j)}|$ corresponding to a particular $k$-partition of the graph $G$, it holds that

$$\mathcal{W}_G^{\gamma=0}(\rho) \leq n - R_k^{\gamma=0} = n - |\overline{E}_{mcm}^{(k)}|. \tag{17}$$

For $0 \leq \gamma \leq 1$, it is easy to see that

$$\mathcal{W}_G^{\gamma}(\rho) = \gamma \mathcal{W}_G^{\gamma=1}(\rho) + (1-\gamma)\mathcal{W}_G^{\gamma=0}(\rho), \tag{18}$$

which is upper bounded by the linear combination of the bounds from Eqs. (15) and (17), giving us

$$\mathcal{W}_G^{\gamma}(\rho) \leq n + \gamma(|E| - |\overline{V}^{(k)}|) - (1-\gamma)|\overline{E}_{mcm}^{(k)}|. \tag{19}$$

Due to the convexity of $\mathcal{W}_G^{\gamma}(\rho)$ in $\rho$, the bound in Eq. (19) holds for all $k$-separable states of this specific tensor product structure $\sigma_k = \sum_i p_i \otimes_{j=1}^{k} |\psi_i^{(j)}\rangle\langle\psi_i^{(j)}|$. This completes the proof of Lemma 1.

In order to have a valid upper bound for all $k$-separable states $\rho_k$, we must minimize the upper bound over all $k$-partitions of $G$, resulting in the first inequality of Eq. (4):

$$\mathcal{W}_G^{\gamma}(\rho) \leq n + \gamma|E| - \min_{\text{all } k-\text{cuts}}\left(\gamma|\overline{V}^{(k)}| + (1-\gamma)|\overline{E}_{mcm}^{(k)}|\right). \tag{20}$$

Finally, it remains to prove the second inequality in Eq. (4). First, by Proposition 2, the subgraph $\overline{G}^{(k)} = (\overline{V}^{(k)}, \overline{E}^{(k)})$ corresponding to any $k$-cut/partition must have at least one vertex in each partition that is connected to a vertex in another partition. Thus, we have $|\overline{V}^{(k)}| \geq k$ for all $k$-cuts with $k \geq 2$. Next, since every connected component of a $k$-cut subgraph $\overline{G}^{(k)}$ has at least two vertices, $\overline{G}^{(k)}$ must have a matching with at least one edge, implying that $|\overline{E}_{mcm}^{(k)}| \geq 1$. Using these two inequalities, we obtain the second inequality of Eq. (4).

## Details of the SDP for incomplete measurements

As mentioned in section "SDP for incomplete measurements", we use SDP to lower bound the experimentally inaccessible terms in Eq. (2), constrained by the expectation values of measurable correlators. The general strategy is to solve the optimization problem:

$$\alpha := \min_{\rho} \sum_{i=1}^{N} |\text{Tr}(A_i\rho)| \tag{21a}$$

subject to $|\text{Tr}(B_j\rho) - b_j| \leq \varepsilon_j \; \forall j \in [J]$, (21b)

$$\text{Tr}(\rho) = 1, \rho \geq 0, \tag{21c}$$

where $A_i$ denote the inaccessible Hermitian operators, $B_j$ denote the measurable Hermitian operators, $b_j$ denote the observed expectation values of $B_j$, and $\varepsilon_j$ denote the uncertainty in measuring $B_j$. Since both the objective function and some of the constraints are nonlinear, we must first linearize the above problem before we can apply standard SDP techniques. A brief review of SDP can be found in Supplementary Note 6.

The linearized version is given by the following (primal) SDP problem:

$$\alpha := \min_{\widetilde{X}} \text{Tr}[(\mathbb{1}_N \oplus \mathbf{0}_d)\widetilde{X}] \tag{22a}$$

subject to $\text{Tr}[(-|i\rangle\langle i| \oplus A_i)\widetilde{X}] \leq 0$, (22b)

$$\text{Tr}[-(|i\rangle\langle i| \oplus A_i)\widetilde{X}] \leq 0, \tag{22c}$$

$$\text{Tr}[(\mathbf{0}_N \oplus B_j)\widetilde{X}] \leq b_j + \varepsilon_j, \tag{22d}$$

$$\text{Tr}[(\mathbf{0}_N \oplus -B_j)\widetilde{X}] \leq -b_j + \varepsilon_j, \tag{22e}$$

$$\text{Tr}[(\mathbf{0}_N \oplus \mathbb{1}_d)\widetilde{X}] = 1, \tag{22f}$$

$$\widetilde{X} \geq 0, \tag{22g}$$

where $\mathbf{0}_d$ denotes a $d \times d$ zero matrix, $\widetilde{X}$ is a $(N + d)$-dimensional positive semi-definite matrix, and $|i\rangle\langle i|$ denotes an $N \times N$ matrix with the $i$-th diagonal entry being 1 and all the remaining entries being 0. The constraints in Eqs. (22b–e) holds for all $i \in [N]$ and $j \in [J]$. The corresponding dual SDP problem can be written as:

$$\beta := \max_{z,\vec{y}} z + \sum_{j=1}^{J}(b_j + \varepsilon_j)y_{2j-1} + (-b_j + \varepsilon_j)y_{2j} \tag{23a}$$

subject to $z\mathbb{1}_d + \sum_{j=1}^{J}(y_{2j-1} - y_{2j})B_j$

$$+ \sum_{i=1}^{N}(y_{2(J+i)-1} - y_{2(J+i)})A_i \leq \mathbf{0}_d, \tag{23b}$$

$$y_{2(J+i)-1} + y_{2(J+i)} \geq -1 \; \forall i \in [N], \tag{23c}$$

$$\vec{y} \leq \vec{0}_{2(N+J)}, z \in \mathbb{R}, \tag{23d}$$

where $\vec{0}_d$ denotes a $d$-dimensional zero vector.

Due to numerical imprecision, the solution for the original optimization problem or the primal SDP problem may be suboptimal, which can lead to an overestimation of the sum of inaccessible terms in Eq. (2), potentially certifying more entanglement than is actually present. The advantage of lower bounding the sum of inaccessible terms by solving the dual SDP problem is that the dual optimum $\beta$ must be less than or equal to the primal optimum $\alpha$ by *weak duality*[87]. Hence, any suboptimal solution to the dual problem obtained from numerical

optimizations must be a valid lower bound to the true minimum of the primal problem $\alpha$. This ensures that we do not overestimate the entanglement when using the numerical solution to the dual problem from numerical solvers as a lower bound for the minimum value of the sum of inaccessible stabilizer terms compatible with all available measurements.

In addition, since there exist vectors $\vec{y}$ and real numbers $z$ that satisfy all the feasibility conditions of the dual problem with strict inequalities (e.g., $\vec{y} \oplus z < \vec{0}_{2(N+J)+1}$ with $y_{2\kappa-1} = y_{2\kappa} \; \forall \; \kappa \in [N+J]$ and $y_{2(J+i)-1} + y_{2(J+i)} > -1 \; \forall \; i \in [N]$), we apply Slater's theorem (see Lemma 3 in Supplementary Note 6) to obtain the following observation.

**Observation 2.** *Strong duality* holds for the above SDP problems (i.e., $\alpha = \beta$).

This observation guarantees that the dual optimal solution $\beta$ is equal to the primal optimal solution $\alpha$, meaning that the solution we get for the dual SDP problem from numerical optimization will be a tight lower bound (as tight as numerical optimization can achieve) of the true minimum of the optimization problem in Eqs. (21a)–(21c).

## Data availability
No data sets were generated or analyzed during the current study.

## Code availability
The code that supports the findings of this study is available in ref. 88.

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

## Acknowledgements

We thank Yuri Minoguchi and Angelika Wiegele for insightful discussions. We also thank Yifan Tang and Zhenhuan Liu for helpful discussions regarding ref. 55. X.D. and K.R. acknowledge support from ETH Zurich and are grateful for the discussions with members of the Quantum Device Lab from ETH Zurich. This research was funded in whole or in part by the Austrian Science Fund (FWF) [https://doi.org/10.55776/P36478]. For open access purposes, the author has applied a CC BY public copyright license to any author-accepted manuscript version arising from this submission. We further acknowledge support from the Austrian Federal Ministry of Education, Science and Research via the Austrian Research Promotion Agency (FFG) through the flagship project FO999897481 (HPQC), the project FO999914030 (MUSIQ), and the project FO999921407 (HDcode) funded by the European Union—Next-GenerationEU, from the European Research Council (Consolidator grant "Cocoquest" 101043705), and the Horizon-Europe research and innovation program under grant agreement No 101070168 (HyperSpace). Additional support is acknowledged from the Canada First Research Excellence Fund.

## Author contributions

N.K.H.L., X.D., M.H., and N.F. conceived the main ideas behind this work. N.K.H.L. discovered and derived all the theoretical results and applied the GME and $k$-inseparability criteria to the simulated data. X.D. and K.R. proposed the experimental setup and simulation conditions. M.H.M.-A. performed all the simulations and data post-processing. All authors discussed the results and contributed to the writing of the final manuscript.

## Competing interests

The authors declare no competing interests.
