## [Transparent Peer Review file · Nature Communications]

Detecting genuine multipartite entanglement in multi-qubit devices with restricted measurements

Corresponding Author: Mr Nicky Kai Hong Li

Version 0:

Reviewer comments:

Reviewer #1

(Remarks to the Author)

The manuscript “Detecting genuine multipartite entanglement in multi-qubit devices with restricted measurements” presents a family of criteria for detecting genuine multipartite entanglement (GME) and k -inseparability in multi-qubit systems under restricted measurement conditions. The study of GME is important for quantum technologies, yet it is notoriously challenging—existing tests often demand non-local measurements, an exponential number of measurements, or even quantum state tomography. The authors propose stabilizer-based graph criteria that only require few-body measurements: the number of measurements scales at most quadratically with system size, and in many cases, involves only constant-weight stabilizers. At its core, the proposed method adopts a fidelity-based approach, linking stabilizer expectation values to fidelities relative to target graph states. The work is complemented by semidefinite programming techniques for handling incomplete data, and supported by both analytical calculations of noise thresholds and numerical simulations of microwave-photonics graph states in superconducting circuits. Overall, the approach is practical and versatile, holding potential to make a valuable contribution to benchmarking multipartite entanglement in realistic experimental platforms.

In general, I support its publication in Nature Communications. However, the manuscript requires substantial revisions before it can be published. My specific comments are as follows:

1. The authors should add comparisons with existing fidelity-based methods. Fidelity-based entanglement detection is already a well-studied field, and the authors must explicitly compare their approach with established fidelity-based methods (e.g., those in Refs. [43, 44]). Such a comparison is essential to clarify the novelty and concrete advantages of their proposal. Additionally, it would be valuable to discuss how the proposed stabilizer-based method relates to fidelity-based approaches for non-stabilizer target states (e.g., symmetric states), thereby expanding the contextual relevance of the work.
2. The manuscript should also compare the proposed method with moment-based GME detection approaches (e.g., PRL 129, 260501), highlighting differences in measurement requirements, computational complexity, and performance under realistic noise conditions.
3. Clarification is needed for the construction of Eq. (2). The paper introduces Eq. (2) as the core object of the new criteria, yet its construction and underlying intuition are not sufficiently explained. A clearer discussion of how the stabilizer combinations are selected—and why this selection yields an effective entanglement witness—would significantly improve readability. For example, the authors should address: why choose stabilizers $\langle S_i \rangle$ and $\langle S_j \rangle$ where i and j are adjacent nodes in the graph?
4. The relationship between graph structure and the choice of parameter γ requires further elaboration. γ plays a crucial role in the proposed criteria, but the physical or mathematical rationale for how its optimal value depends on the graph structure of the target state remains underdeveloped. The authors should expand on this relationship and ideally provide additional examples (e.g., for different graph topologies like linear chains, star graphs, or complete graphs) to illustrate how γ is tuned.

The writing and presentation of the manuscript need improvement. While the work is technically rigorous, its exposition is relatively unclear, with multiple instances of notation abuse and imprecision that obscure key ideas. For example:

- The symbol $\langle \dots \rangle$ is used ambiguously, sometimes denoting expectation values and other times denoting generated groups;
- In Eq. (1), the expression fails to clearly indicate that the partition of subsystems may depend on the index i ;
- The letter X —conventionally used to denote the Pauli X operator—is also used to represent a generic operator, leading to

potential confusion;

- The statement “For any connected graph, our criteria only require measuring J out of the total of ...” is vague (e.g., J is not defined or contextualized) and needs clarification.
- The manuscript would benefit from a careful, thorough revision to enhance the precision of notation, the clarity of explanations, and overall consistency throughout.

(Remarks on code availability)

Reviewer #2

(Remarks to the Author)

The article is solid science on the topic of detecting genuine multipartite entanglement and non- k -separability in graph states. The key contribution is that the authors manage to reduce the number of measurements from $O(2^n)$ to $O(n^2)$, in some cases even to $O(n)$. The work is a nice strengthening of previous results, but it is also restricted to stabilizer states for which already a lot is known.

The key contribution is the reduction in measurements for a common class of states.

In my view this does not pass the threshold for Nature Communications, but could be convinced if the insight behind the results translates to other settings.

Major comments:

The article needs more insight/intuition on why these criteria and why matchings / cuts show up. Defining a witness as in Eq. [2] or Theorem 1 is fine, but the conceptual contribution or insight is not well highlighted. In Proposition 1 & 2, what are novel insights / differences to previous similar bounds by the authors?

Minor:

The optimization in (8a) is not very scalable and can be performed only for small system sizes.

(Remarks on code availability)

Reviewer #3

(Remarks to the Author)

The authors study practical (in the sense of requiring a small number of stabilizer measurements, which only act on a small number of qubits at the same time) entanglement witnesses, which can certify non only GME, but also general k -separability. The entanglement witness is tailored to a given graph state, but can also certify the GME of interesting non-stabilizer states. The authors contributions are noteworthy from both the theoretical point of view, but they also took great care to ensure their results are practically applicable. They do so by suggesting experimental implementations, simulating them in (what I presume are) realistic experimental situations, and by considering SDP to further ease the experimental implementation.

Overall, I think the paper is very well-written, and I would expect that this could naturally lead to experiments in the near-term. The introduction in particular was enjoyable to read, providing a clear motivation for this work, and furthermore also provided enough of a literature review to put this work into context.

I do, however, have some comments that should be addressed. My main concern is the following. The motivation for minimizing the m -body correlators is that it is hard to perform them. However, to estimate the expectation value of an m -body Pauli string, it suffices to perform single-qubit Pauli measurements and record the parity of all measurements. Single-qubit Pauli measurements are clearly easier to implement than larger weight Pauli strings (this is essentially one of the biggest motivations for studying this problem, according to the authors). Unless I am missing something, this severely limits the motivation for minimizing the weight of Pauli measurements performed.

There are several reasons I could think of why such single-qubit Pauli measurements might be bad, however. Clearly for each measurement one can only estimate the expectation value of Pauli measurements that are locally commutative (i.e. subsets of stabilizers whose restrictions to individual qubits pairwise commute), which would correspond for example to the ‘vertex stabilizers’ that form independent sets in the graph. After such a measurement, the state would need to be prepared again (note that this also motivates the concept of local measurement settings studied previously). Here I could imagine the authors’ approach could provide some gain, since the m -body stabilizers S_i can be measured in succession, without having to recreate the state. At the same time, if i and j are connected, then after measuring S_i and S_j there is no use in measuring $S_i * S_j$, since measuring S_i and S_j projects the state into a well-defined eigenstate of their product. So this would require re-preparing the state anyway when measuring the $S_i * S_j$ stabilizers. Furthermore, it is not clear whether performing some of these multi-body measurements in succession (without having to reprepare the state) is actually beneficial in an experiment, since this incurs extra decoherence because the state is idling and one measures larger body correlators.

It might be that the above concern is easily addressed, since I'm not an expert on (experimental considerations) of entanglement witnesses. If it is, it would be good to highlight in the paper what the experimental drawbacks would be for using single-qubit Pauli measurements, as opposed to measuring the full Pauli strings directly/the procedure that the authors propose.

Minor concerns:

It took some time to understand what was meant in theorem 1 under eq 4. It would help to be more precise here, and it would also help (but is not necessary) to have Fig. 1 on the same page.

It would be good to mention the notion of locally equivalent graph states after remark 1. In particular that one optimize the graph used in the entanglement witness over all graphs in the local equivalence class. For example, all graphs in Fig. A.2 are locally equivalent (if we include qubit permutations). Furthermore, the second bound in eq (3) motivates the study of optimizing the number of edges in graph states up to local complementations, which can be done using the results in for example 2506.00292.

I think remark 2 could be sharpened, by first saying that the criterion based on G is tailored to $|G\rangle$, then saying that it achieves $n + \gamma |E|$ for $|G\rangle \prec G$, and then that it can still apply to other non-stabilizer states.

Proposition 1 has the definition of the anti-commutator, while it is already used in lemma 2.

I would change the word 'define' in lemma 2 by 'let'.

"As the stabilizer term $|\langle S_b \rangle|$ of $b \in V_{\text{match}}^k$ is* already paired (...)"

In A.1 the authors highlight how other witnesses do not detect between different levels of k -inseparability. I feel this could be stressed more in the main text as a novel result.

(Remarks on code availability)

The code does not include the calculations for the expectation values from the density matrices, as far as I can tell.

Version 1:

Reviewer comments:

Reviewer #1

(Remarks to the Author)

The concerns and issues raised in the previous round of review have been carefully and adequately addressed by the authors in the revised manuscript. The current version demonstrates improved clarity and correctness. Therefore, I recommend this work for publication.

Reviewer #2

(Remarks to the Author)

The authors addressed my major and minor concerns. It reads much better now with a clear explanation of the key concept. However, due to the somewhat incremental nature (using locally commuting stabilizer operators with Cauchy Schwarz), I am not 100% sure whether the paper represents an advance of significance to specialists within the field.

Reviewer #3

(Remarks to the Author)

The authors have addressed all my questions, in particular my concern regarding minimizing the weight of the stabilizers is sufficiently addressed. As such, I now recommend acceptance in Nature Communications.

Response to Reviewer 1

Thank you for reviewing our manuscript for *Nature Communications*. We appreciate the time and effort that you put into the review process and providing constructive feedback on our manuscript. We addressed all comments in the revised version of our manuscript. All changes are highlighted in purple texts in the attached manuscript. Please find our in-line response below. Note that all page and section numbers refer to the attached manuscript and supplementary information in PDF **with purple texts** that highlight the amendments.

With best regards,

Nicky Kai Hong Li (on behalf of all authors)

Inline Response:

Reviewer #1 (Remarks to the Author):

The manuscript “Detecting genuine multipartite entanglement in multi-qubit devices with restricted measurements” presents a family of criteria for detecting genuine multipartite entanglement (GME) and k -inseparability in multi-qubit systems under restricted measurement conditions. The study of GME is important for quantum technologies, yet it is notoriously challenging—existing tests often demand non-local measurements, an exponential number of measurements, or even quantum state tomography. The authors propose stabilizer-based graph criteria that only require few-body measurements: the number of measurements scales at most quadratically with system size, and in many cases, involves only constant-weight stabilizers. At its core, the proposed method adopts a fidelity-based approach, linking stabilizer expectation values to fidelities relative to target graph states. The work is complemented by semidefinite programming techniques for handling incomplete data, and supported by both analytical calculations of noise thresholds and numerical simulations of microwave-photonic graph states in superconducting circuits. Overall, the approach is practical and versatile, holding potential to make a valuable contribution to benchmarking multipartite entanglement in realistic experimental platforms.

In general, I support its publication in *Nature Communications*. However, the manuscript requires substantial revisions before it can be published. My specific comments are as follows:

General Response: We sincerely thank the referee for the thoughtful and encouraging assessment of our work and for the constructive comments and questions. In addition to addressing the specific points raised, we would like to highlight **two substantial improvements** made in the revised manuscript.

First, we have clarified and emphasized an important practical aspect of our criteria: optimization under local Clifford (LC) equivalence of graph states. As explained now in the **main text (after Remark 1)**, LC-equivalent graph states may differ in maximum degree or number of edges, quantities that directly determine stabilizer weights and the number of measurements in our criteria. Choosing an LC-representative with a smaller maximum degree or number of edges can therefore significantly reduce the experimental effort required to evaluate our criteria. This perspective is incorporated into the **updated abstract and Introduction (third-to-last paragraph)**, formalized in a **new Remark 2**, and **illustrated explicitly in Appendix A.V**.

Second, we have **substantially expanded the Discussion** section to articulate the broader conceptual scope and impact of our results. Our criteria admit an interpretation in terms of a Pauli-local Hamiltonian whose structure resembles families of models studied in the context of symmetry-protected topological (SPT) phases [86, 87]. Constructions such as “pivoting” [87], in

which graph-local unitaries transform simple reference models (e.g., classical Ising models) into stabilizer-type Hamiltonians, yield a class of models with analytically tractable ground and low-energy excited states, and provide a useful conceptual parallel to the Hamiltonian form underlying our criteria. While originating from different motivations, our approach suggests a complementary direction by offering a theoretically tractable method for probing entanglement properties of highly excited states, a regime that is often challenging for conventional numerical techniques. Moreover, the relevant observables are local, in the sense that they involve only a few qubits for many important interaction graphs (e.g., 1D and 2D clusters), and are therefore accessible in current experimental platforms.

Importantly, this Hamiltonian perspective shows that our method yields experimentally friendly GME/ k -inseparability criteria based on mean-energy measurements, quantities that are typically accessible in quantum many-body experiments. This substantially broadens the applicability of our approach beyond graph states and stabilizer architectures and provides a concrete pathway for deploying our criteria in diverse physical settings governed by local Hamiltonians. Looking ahead, the structural similarity to many-body Hamiltonians points toward a broader framework in which entanglement witnesses or detection criteria may be systematically derived by combining insights from many-body physics with multipartite entanglement theory.

We believe these additions significantly strengthen the scope and practical relevance of our results, and demonstrate that the underlying insights of our method extend beyond stabilizer states and to other physical settings.

1. The authors should add comparisons with existing fidelity-based methods. Fidelity-based entanglement detection is already a well-studied field, and the authors must explicitly compare their approach with established fidelity-based methods (e.g., those in Refs. [43, 44]). Such a comparison is essential to clarify the novelty and concrete advantages of their proposal. Additionally, it would be valuable to discuss how the proposed stabilizer-based method relates to fidelity-based approaches for non-stabilizer target states (e.g., symmetric states), thereby expanding the contextual relevance of the work.

Response: We thank the reviewer for this valuable suggestion and we agree that comparisons with established fidelity-based methods will help to place our contribution in an appropriate context.

In response, we have **added a direct comparison between our criteria and the GME/ k -inseparability witnesses of [npj Quantum Inf. 5, 83 (2019)]** (now cited as Ref. [47]) **in Tables I–IV and in Sec. II E.** These tables showcase the performance of several witnesses/criteria on simulated noisy graph states. While the witnesses in Ref. [47] perform slightly better than our criteria, they generally require measuring at least $O(2^{n/c})$ stabilizers with a maximum degree of up to $O(n)$ (where c is the chromatic number of the graph). Such measurements are not experimentally feasible in the platforms that we consider when n becomes large, due to connectivity constraints or rapidly increasing measurement noise.

Let us also note that, to the best of our knowledge, other existing stabilizer/fidelity-based GME witnesses do not certify general k -inseparability (except for full separability). This is why we believe that the addition of the comparison with Ref. [47] already gives a representative and comprehensive comparison to the relevant fidelity-based approaches. In particular, Tables I–IV now contrast our criteria with two extremal classes of GME witnesses: (1) the witness in Eq. (45) of Ref. [46], which fulfills the same measurement restrictions as our method and uses the minimum number of stabilizers; and (2) the witness based on fidelity > 0.5 in Ref. [45], which requires measuring all 2^n stabilizers.

2. The manuscript should also compare the proposed method with moment-based GME detection approaches (e.g., PRL 129, 260501), highlighting differences in measurement requirements, computational complexity, and performance under realistic noise conditions.

Response: We also thank the reviewer for this helpful suggestion and for pointing us to this valuable reference. We have now mentioned this method in the **Introduction (4th last paragraph)** and **extended the review section in Appendix A.I** to summarize the measurement requirements and computational complexity requirements of [PRL 129, 260501] (now cited as Ref. [55]).

3. Clarification is needed for the construction of Eq. (2). The paper introduces Eq. (2) as the core object of the new criteria, yet its construction and underlying intuition are not sufficiently explained. A clearer discussion of how the stabilizer combinations are selected—and why this selection yields an effective entanglement witness—would significantly improve readability. For example, the authors should address: why choose stabilizers $\langle S_i \rangle$ and $\langle S_j \rangle$ where i and j are adjacent nodes in the graph?

Response: We thank the referee for this useful feedback. We have substantially expanded the discussion surrounding Eq. (2) to make its construction and intuition clearer.

The key idea behind our method is that the entire set of stabilizers is not actually needed to obtain good bounds for certifying entanglement. Once the union of all the edges corresponding to the stabilizers form a connected graph, a polynomially small subset of local stabilizers, together with the positivity of quantum states, is already sufficient to certify important properties of many-body quantum states.

More precisely, the choice of stabilizer subset stems from the use of the *anticommutativity inequality* (Lemma 2 and Proposition 1 in Methods) and the fact that Pauli matrices anticommute, which together lead to the analytic bounds in Eqs. (4) and (8). These properties ensure that whenever two qubits a and b in ρ correspond to adjacent vertices in the underlying graph and belong to different groups of a given k -partition of the n qubits, every k -product state of that k -partition satisfies $|\langle S_a \rangle| + |\langle S_b \rangle| + |\langle S_a S_b \rangle| \leq 1$, $|\langle S_a \rangle| + |\langle S_a S_b \rangle| \leq 1$, and $|\langle S_a \rangle| + |\langle S_b \rangle| \leq 1$. These constraints arise because, after factorization across the partition, the operators acting on each subsystem form anticommuting triples or pairs whose squared expectation values sum to at most one. Alternatively or additionally, one may also derive bounds by leveraging the positivity constraint via a semidefinite programming (SDP) approach, as done in Sec. II C.

To illustrate this more concretely, consider the example of a 5-qubit graph in Fig. 1 of the manuscript, and the partition of the 5 qubits/vertices into three sets, $\{1,4,5\}$, $\{2\}$, and $\{3\}$. Since Eq. (2) contains sums over stabilizers for all vertices and products of stabilizers corresponding to all pairs of neighboring vertices, the quantity in Eq. (2) contains (amongst others) terms such as $|\langle S_1 \rangle| = |\langle X_1 Z_2 Z_3 Z_5 \rangle|$, $|\langle S_2 \rangle| = |\langle X_2 Z_1 Z_3 \rangle|$, and $|\langle S_1 S_2 \rangle| = |\langle Y_1 Y_2 Z_5 \rangle|$, where the fact that vertices 1 and 2 are connected by an edge means that the Pauli- Y terms appear as products of X and Z operators for both qubits 1 and 2. For any state that is separable with respect to the chosen partition in Fig. 1, these stabilizer expectation values factorize as $|\langle S_1 \rangle| = |\langle X_1 Z_5 \rangle| |\langle Z_2 Z_3 \rangle|$, $|\langle S_2 \rangle| = |\langle Z_1 \rangle| |\langle X_2 Z_3 \rangle|$, and $|\langle S_1 S_2 \rangle| = |\langle Y_1 Z_5 \rangle| |\langle Y_2 \rangle|$. Hence, the sum of these three terms can be bounded via the Cauchy-Schwarz inequality,

$$\begin{aligned} |\langle S_1 \rangle| + |\langle S_2 \rangle| + |\langle S_1 S_2 \rangle| &= |\langle X_1 Z_5 \rangle| |\langle Z_2 Z_3 \rangle| + |\langle Z_1 \rangle| |\langle X_2 Z_3 \rangle| + |\langle Y_1 Z_5 \rangle| |\langle Y_2 \rangle| \\ &\leq (|\langle X_1 Z_5 \rangle|^2 + |\langle Z_1 \rangle|^2 + |\langle Y_1 Z_5 \rangle|^2)^{1/2} (|\langle X_2 Z_3 \rangle|^2 + |\langle Z_2 Z_3 \rangle|^2 + |\langle Y_2 \rangle|^2)^{1/2}. \end{aligned}$$

For the sums of squared expectation values under each square root, we can use the anticommutativity bound (Lemma 2 and Proposition 1 in Methods) to bound them by a constant (1 in this case): for each side of the partition, the Pauli operators $\{X, Y, Z\}$ (tensored with spectator Pauli- Z 's) form a maximally anticommuting triple. We can also make similar arguments for any other pair of neighboring vertices that is “separated” by the selected partitioning, and then repeat this procedure for all possible k -partitions, thus arriving at a bound for all k -separable states.

A key structural point is that this argument applies *only* when the two vertices are connected by an edge and separated by the partition. Thus, by including all vertex stabilizers and all pairs of stabilizers for all edges, we (i) maximize the number of terms that contribute to the left-hand side of the bound, while (ii) ensuring that we still have a maximal set of anticommuting operators under each square root after the factorization.

In contrast, states that are inseparable across this partition can exceed these bounds since the three sums at the beginning can reach a maximum of 3, 2 and 2, respectively (e.g., when $\rho = |G\rangle\langle G|$). These structural dependence on the graph adjacencies, arising specifically from our chosen subset of stabilizers, explain why matchings and cuts naturally enter the formulation of the optimal bounds and, crucially, why our approach can certify not only GME but also general k -inseparability.

To address your comment, we have now **(i) added some text to clarify and guide this intuition after Eq. (2), (ii) inserted a further clarifying remark immediately before Theorem 1, and (iii) substantially revised Fig. 1 and its caption** to better illustrate the corresponding graph-theoretic objects and intuition.

4. The relationship between graph structure and the choice of parameter γ requires further elaboration. γ plays a crucial role in the proposed criteria, but the physical or mathematical rationale for how its optimal value depends on the graph structure of the target state remains underdeveloped. The authors should expand on this relationship and ideally provide additional examples (e.g., for different graph topologies like linear chains, star graphs, or complete graphs) to illustrate how γ is tuned.

Response: We agree that this requires further elaboration. In general, the optimal choice of the parameter γ for achieving maximum violation of the inequality depends on two factors: (i) the measured values of the two summation terms in Eq. (2), and (ii) the number of edges and the reduction term R_k^γ , which behave differently for different underlying graphs. Since factor (i) can vary a lot across different experiments, the optimal γ may vary from experiment to experiment even for the same underlying graph. Therefore, there is generally no analytical strategy for finding the optimal γ . The most straightforward strategy is to measure W_G^γ then perform the maximization as described in Eq. (6) numerically, which is how we obtained the optimal γ for the simulated data in Tables I–IV and Fig. 4. Given that there’s only one parameter to be optimized, the maximization is typically straightforward. **We have now added a short note under Eq. (6) to elaborate on this point.**

For the idealized examples of graph states mixed with white noise [see the Cthulhu-graph example (see around Observation 1 and Appendix A.III) and Appendix A.VII], the two summation terms in Eq. (2) have a simple relationship as described in Eq. (A.16), so the optimal γ depends entirely on factor (ii) which is determined by the underlying graph’s structure. In the white-noise added scenario, we observed that the optimal γ for all 1D cluster, ring, star, and tree graphs is 1 for certifying k -inseparability for all $2 \leq k \leq n$, whereas for 2D cluster and complete graphs, the optimal γ can be either 0 or 1 depending on k and the size of the graph. In particular, the optimal γ is 0 for n -vertex complete graphs where $n \geq 4$ and $3 \leq k \leq n$. The details can be found in **Appendix A.VII**, where

we have now **explicitly stated the optimal γ (denoted by γ^*) for each of the above cases** in the new version. However, we must admit that the reason why the optimal γ can switch between 0 and 1 alternatively for different k in white-noise added 2D cluster states is still unclear to us, and could be an interesting future research direction.

To understand conceptually the interesting situation where the optimal γ (e.g., the Cthulhu-graph example) is strictly between 0 and 1, we observe that Cthulhu graphs are composed of a complete graph (as its “head”) and a star graph (as its “tentacles”), of which the corresponding noisy graph states have optimal γ being 0 and 1, respectively, when certifying their k -inseparability for $n \geq 4$ and $3 \leq k \leq n$. Therefore, it is not hard to believe that the optimal γ for certifying k -inseparability of noisy Cthulhu-graph states interpolate the extremal values 0 and 1. However, to show that the precise analytical conditions for this to happen is more subtle, which we showed around Observation 1 and in Appendix A.III. To provide more intuition on how to understand this special situation, **we added a short note after Eq. (7) and modified Fig. 2 (and its caption).**

The writing and presentation of the manuscript need improvement. While the work is technically rigorous, its exposition is relatively unclear, with multiple instances of notation abuse and imprecision that obscure key ideas. For example:

- The symbol $\langle \dots \rangle$ is used ambiguously, sometimes denoting expectation values and other times denoting generated groups;

Response: We agree that this notation can be confusing for the readers. Therefore, we have now **deleted the text** “i.e., $\$ \operatorname{Stab}(\ket{G}) = \langle S_i | i \in V \rangle$ ” **from the 2nd paragraph of Sec. IIA** since it doesn’t add any extra useful information compared to the previous line that states that these stabilizers S_i generate the full stabilizer group, and we are not using this notation anymore in the remaining article.

- In Eq. (1), the expression fails to clearly indicate that the partition of subsystems may depend on the index i ;

Response: Thanks for pointing this out. We updated the notation in Eq. (1) and added some clarification right after.

- The letter X —conventionally used to denote the Pauli X operator—is also used to represent a generic operator, leading to potential confusion;

Response: Thanks again. We replaced the letters X and Y with A and B to denote generic operators after Eq. (2) and in Proposition 1. In Eqs. (21) and Appendices A.VI & A.X, we replaced X and Y with \tilde{X} and \tilde{Y} .

- The statement “For any connected graph, our criteria only require measuring J out of the total of ...” is vague (e.g., J is not defined or contextualized) and needs clarification.

Response: We moved the definitions (range of allowed values) of J and M earlier (see the paragraph below Eq. (2)) to avoid confusion.

- The manuscript would benefit from a careful, thorough revision to enhance the precision of notation, the clarity of explanations, and overall consistency throughout.

Response: Thank you very much for going through the notations of our manuscript thoroughly. We have gone through the paper, clarified any confusing notations, and modified any unclear sentences to improve readability.

Response to Reviewer 2

Dear Referee,

Thank you for reviewing our manuscript for *Nature Communications*. We appreciate the time and effort that you put into the review process and providing constructive feedback on our manuscript. We addressed all comments in the revised version of our manuscript. All changes are highlighted in purple texts in the attached manuscript. Please find our inline response below. Note that all page and section numbers refer to the attached manuscript and supplementary information in PDF **with purple texts** that highlight the amendments.

With best regards,

Nicky Kai Hong Li (on behalf of all authors)

Inline Response:

Reviewer #2 (Remarks to the Author):

The article is solid science on the topic of detecting genuine multipartite entanglement and non- k -separability in graph states. The key contribution is that the authors manage to reduce the number of measurements from $O(2^n)$ to $O(n^2)$, in some cases even to $O(n)$. The work is a nice strengthening of previous results, but it is also restricted to stabilizer states for which already a lot is known.

The key contribution is the reduction in measurements for a common class of states. In my view this does not pass the threshold for Nature Communications, but could be convinced if the insight behind the results translates to other settings.

Response: We gratefully acknowledge the referee's careful review and positive assessment of our work. The referee raises an important question regarding the generality of our framework and the relevance of our insights to other settings.

First, we fully agree that broad applicability is an important consideration. We would like to clarify that our framework is not restricted to stabilizer states—as demonstrated in Appendix A.VII(d), our method can certify GME/ k -inseparability in non-stabilizer families such as states LU-equivalent to noisy Dicke states. In the revised manuscript, we have put more emphasis on this point (see the **second-to-last paragraph of the Introduction and the 2nd paragraph of the Discussion**).

Second, we have **substantially expanded the Discussion section** in the revised manuscript to highlight the broader impact and scope of our results. In particular, we point out an important conceptual connection between our work and quantum many-body physics, where entanglement has become a much-considered tool for classifying phases of matter. Our criterion admits an interpretation as a Hamiltonian whose Pauli-local structure resembles families of models recently studied in the context of symmetry-protected topological (SPT) phases [86,87]. This type of Hamiltonian can be related to a construction known as “pivoting” [87], in which graph-local unitaries transform simple reference models (e.g., classical Ising models) into stabilizer-type Hamiltonians. Such an approach yields a class of models with analytically tractable ground and low-energy excited states. Our work suggests a potential path to go beyond this paradigm by providing theoretically tractable criteria for studying entanglement properties of highly excited states—a regime that is often challenging for conventional numerical techniques. The observables required for our method are “local” in the sense that it involves only a few qubits for many important classes of interaction graphs (e.g., 1D & 2D clusters), making them accessible in current experimental platforms. While our criteria originate from different motivations, the structural similarity to many-body Hamiltonians points toward a broader

framework in which entanglement witnesses or detection criteria may be systematically derived by combining insights from many-body physics with multipartite entanglement theory.

Importantly, this Hamiltonian perspective shows that our method yields experimentally friendly GME/ k -inseparability criteria based on mean-energy measurements—quantities that are typically accessible in quantum many-body experiments. This significantly broadens the reach of our approach beyond graph states and stabilizer architectures and provides a concrete pathway for applying our criteria in diverse physical settings governed by local Hamiltonians.

We believe this generalization significantly strengthens the scope and relevance of our results and demonstrates that the underlying insights of our method extend beyond stabilizer states and to other physical settings.

Major comments:

The article needs more insight/intuition on why these criteria and why matchings / cuts show up. Defining a witness as in Eq. [2] or Theorem 1 is fine, but the conceptual contribution or insight is not well highlighted. In Proposition 1 & 2, what are novel insights / differences to previous similar bounds by the authors?

Response:

We thank the referee for raising this important point. We agree that the intuition behind our choice of stabilizer subset and the appearance of matchings/cuts can be better emphasized. We have therefore expanded the discussion around Eq. (2) and Theorem 1 to clarify the conceptual origin of our criteria.

The key idea is that the entire set of stabilizers is not actually needed to obtain good bounds for certifying entanglement. Once the union of all the edges corresponding to the stabilizers form a connected graph, a polynomially small subset of local stabilizers, together with the positivity of quantum states, is already sufficient to certify important properties of many-body quantum states.

More precisely, the main tools we use are the *anticommutativity inequality* (Lemma 2 and Proposition 1 in Methods) and the anticommuting structure of Pauli operators. These two ingredients jointly yield the analytic bounds in Eqs. (4) and (8). In particular, whenever two qubits a and b in ρ correspond to adjacent vertices in the underlying graph and belong to different groups of a given k -partition of the n qubits, every k -product state of that k -partition necessarily satisfies $|\langle S_a \rangle| + |\langle S_b \rangle| + |\langle S_a S_b \rangle| \leq 1$, $|\langle S_a \rangle| + |\langle S_a S_b \rangle| \leq 1$ and $|\langle S_a \rangle| + |\langle S_b \rangle| \leq 1$. These constraints arise because, after factorization across the partition, the operators acting on each subsystem form anticommuting triples or pairs whose squared expectation values sum to at most one. Alternatively or additionally, one may also derive bounds by leveraging the positivity constraint via a semidefinite programming (SDP) approach, as done in Sec. II C.

To illustrate this more concretely, consider the example of a 5-qubit graph in Fig. 1 of the manuscript, and the partition of the 5 qubits/vertices into three sets, $\{1,4,5\}$, $\{2\}$, and $\{3\}$. Since Eq. (2) contains sums over stabilizers for all vertices and products of stabilizers corresponding to all pairs of neighboring vertices, the quantity in Eq. (2) contains (amongst others) terms such as $|\langle S_1 \rangle| = |\langle X_1 Z_2 Z_3 Z_5 \rangle|$, $|\langle S_2 \rangle| = |\langle X_2 Z_1 Z_3 \rangle|$, and $|\langle S_1 S_2 \rangle| = |\langle Y_1 Y_2 Z_5 \rangle|$, where the fact that vertices 1 and 2 are connected by an edge means that the Pauli- Y terms appear as products of X and Z operators for both qubits 1 and 2. For any state that is separable with respect to the chosen partition in Fig. 1, these stabilizer expectation values factorize as $|\langle S_1 \rangle| = |\langle X_1 Z_5 \rangle| |\langle Z_2 Z_3 \rangle|$,

$|\langle S_2 \rangle| = |\langle Z_1 \rangle| |\langle X_2 Z_3 \rangle|$, and $|\langle S_1 S_2 \rangle| = |\langle Y_1 Z_5 \rangle| |\langle Y_2 \rangle|$. Hence, the sum of these three terms can be bounded via the Cauchy-Schwarz inequality,

$$\begin{aligned} |\langle S_1 \rangle| + |\langle S_2 \rangle| + |\langle S_1 S_2 \rangle| &= |\langle X_1 Z_5 \rangle| |\langle Z_2 Z_3 \rangle| + |\langle Z_1 \rangle| |\langle X_2 Z_3 \rangle| + |\langle Y_1 Z_5 \rangle| |\langle Y_2 \rangle| \\ &\leq (|\langle X_1 Z_5 \rangle|^2 + |\langle Z_1 \rangle|^2 + |\langle Y_1 Z_5 \rangle|^2)^{1/2} (|\langle X_2 Z_3 \rangle|^2 + |\langle Z_2 Z_3 \rangle|^2 + |\langle Y_2 \rangle|^2)^{1/2}. \end{aligned}$$

For the sums of squared expectation values under each square root, we can use the anticommutativity bound (Lemma 2 and Proposition 1 in Methods) to bound them by a constant (1 in this case): for each side of the partition, the Pauli operators $\{X, Y, Z\}$ (tensored with spectator Pauli- Z 's) form a maximally anticommuting triple. We can also make similar arguments for any other pair of neighboring vertices that is “separated” by the selected partitioning, and then repeat this procedure for all possible k -partitions, thus arriving at a bound for all k -separable states.

A key structural point is that this argument applies only when the two vertices are connected by an edge and separated by the partition. Thus, by including all vertex stabilizers and all pairs of stabilizers for all edges, we (i) maximize the number of terms that contribute to the left-hand side of the bound, while (ii) ensuring that we still have a maximal set of anticommuting operators under each square root after the factorization.

In contrast, states that are inseparable across this partition can exceed these bounds since the three sums at the beginning can reach a maximum of 3, 2 and 2, respectively (e.g., when $\rho = |G\rangle\langle G|$). These structural dependence on the graph adjacencies, arising specifically from our chosen subset of stabilizers, explain why matchings and cuts naturally enter the formulation of the optimal bounds and, crucially, why our approach can certify not only GME but also general k -inseparability, which most conventional stabilizer-based criteria cannot.

To address your comment, we have now **(i) added this clarification after Eq. (2)**, **(ii) inserted a further clarifying remark immediately before Theorem 1**, and **(iii) substantially revised Fig. 1 and its caption** to better illustrate the corresponding graph-theoretic objects and intuition.

Regarding the novelty of Propositions 1 and 2: although the anticommutativity inequality of Lemma 2 originates from Ref. [88], its previous use in multipartite entanglement detection has been limited. Earlier applications targeted only GME detection (but not for the more general k -inseparability) and relied only on a restricted subset of two-body Pauli observables (i.e., $X^{\otimes 2}$, $Y^{\otimes 2}$, and $Z^{\otimes 2}$) as in Ref. [1]. There the witnesses are constructed from average fidelities to Bell states, and therefore require only expectation values of pairs of Pauli operators. However, for increasing numbers of qubits, this construction led to a rapid combinatorial growth in the number of terms. While these witnesses could certify GME with the data available in the experiment of Ref. [1], they were insensitive to detect k -inseparability for general values of k (e.g., distinguish between 2- and 3-separability for 4 qubits), and were ineffective for high-fidelity connected graph states. Moreover, it remains unclear if they could, even in principle, detect any GME states beyond 5 qubits.

In contrast, the present work demonstrates several genuinely new features. **First**, we provide a systematic method for constructing witnesses that can detect more complicated entanglement structures using correlations of orders higher than two (between more than two qubits). These correlators involve sets of qubits determined by the local connectivity of the underlying graph, and hence the measurement complexity scales with meaningful graph-theoretic structure rather than with the exponentially growing pairwise combinatorics. **Second**, our approach leads to criteria that are not just systematically constructible but can also efficiently detect both GME and non-GME k -inseparability across a broad class of states (including non-stabilizer examples), even under realistic noise. **Finally**, our approach constitutes the first systematic use of anticommutativity relations to

generate families of multipartite entanglement criteria tailored to stabilizer and graph states, thereby advancing beyond previous bounds both conceptually and operationally.

Minor:

The optimization in (8a) is not very scalable and can be performed only for small system sizes.

Response: Thank you for this comment. In general, using the most stable and commonly used SDP solver MOSEK (which uses the interior-point method) in MATLAB, we can solve the SDP problem that involves up to 12 qubits efficiently on a standard laptop. This means we can readily apply our method to all graph states with $\max_{(i,j) \in E} [d(i) + d(j)] \leq 12$, which include many graph states of practical importance in quantum information science, such as all 1- to 3-dimensional cluster states [20], all ring-graph states [42], and many tree-graph states [43, 44]. Beyond 12 qubits, while solving the SDP may become intractable with conventional interior-point methods, alternative methods with better scalability [80], such as augmented Lagrangian methods, can be employed for larger problems, although the development of more stable software implementations is still required. We have added this comment in Sec. II C.

General Response: In addition to addressing your specific comments, we would like to bring to your attention that we have added to the manuscript an important practical aspect of our criteria: their optimization under local Clifford (LC) equivalence of graph states. We now **explain in the main text (after Remark 1)** that LC operations generate equivalence classes of graph states whose members may differ in their maximum degrees or numbers of edges. Since these graph-theoretic properties directly determine the stabilizer weights and the total number of terms appearing in our criteria, choosing an LC-equivalent representative with a smaller maximum degree or fewer edges can significantly reduce the experimental efforts required to evaluate our criteria. We have **updated the abstract and the introduction (third-to-last paragraph)** accordingly, and have **highlighted this optimization strategy in the newly added Remark 2**. We further illustrate this idea with an **explicit example in Appendix A.V**, where graph-local complementations are used to reduce both the maximum stabilizer weight and the number of stabilizers that need to be measured. We also point readers to recent methods for identifying minimum-edge representatives within an LC-equivalence class. We believe that these additions further strengthen the practical relevance of our framework.

Response to Reviewer 3

Dear Referee,

Thank you for reviewing our manuscript for *Nature Communications*. We appreciate the time and effort that you put into the review process and providing constructive feedback on our manuscript. We addressed all comments in the revised version of our manuscript. All changes are highlighted in purple texts in the attached manuscript. Please find our inline response below. Note that all page and section numbers refer to the attached manuscript and supplementary information in PDF **with purple texts** that highlight the amendments.

With best regards,

Nicky Kai Hong Li (on behalf of all authors)

Inline Response:

Reviewer #3 (Remarks to the Author):

The authors study practical (in the sense of requiring a small number of stabilizer measurements, which only act on a small number of qubits at the same time) entanglement witnesses, which can certify non only GME, but also general k -separability. The entanglement witness is tailored to a given graph state, but can also certify the GME of interesting non-stabilizer states. The authors contributions are noteworthy from both the theoretical point of view, but they also took great care to ensure their results are practically applicable. They do so by suggesting experimental implementations, simulating them in (what I presume are) realistic experimental situations, and by considering SDP to further ease the experimental implementation.

Overall, I think the paper is very well-written, and I would expect that this could naturally lead to experiments in the near-term. The introduction in particular was enjoyable to read, providing a clear motivation for this work, and furthermore also provided enough of a literature review to put this work into context.

General Response: We sincerely thank the referee for the thoughtful and encouraging assessment of our work and for the constructive comments and questions. In addition to addressing the specific points raised, we would like to highlight an **improvement** made in the revised manuscript.

We have **substantially expanded the Discussion** section to articulate the broader conceptual scope and impact of our results. Our criteria admit an interpretation in terms of a Pauli-local Hamiltonian whose structure resembles families of models studied in the context of symmetry-protected topological (SPT) phases [86, 87]. Constructions such as “pivoting” [87], in which graph-local unitaries transform simple reference models (e.g., classical Ising models) into stabilizer-type Hamiltonians, yield a class of models with analytically tractable ground and low-energy excited states, and provide a useful conceptual parallel to the Hamiltonian form underlying our criteria. While originating from different motivations, our approach suggests a complementary direction by offering a theoretically tractable method for probing entanglement properties of highly excited states, a regime that is often challenging for conventional numerical techniques. Moreover, the relevant observables are local, in the sense that they involve only a few qubits for many important interaction graphs (e.g., 1D and 2D clusters), and are therefore accessible in current experimental platforms.

Importantly, this Hamiltonian perspective shows that our method yields experimentally friendly GME/ k -inseparability criteria based on mean-energy measurements, quantities that are typically accessible in quantum many-body experiments. This substantially broadens the applicability of our approach

beyond graph states and stabilizer architectures and provides a concrete pathway for deploying our criteria in diverse physical settings governed by local Hamiltonians. Looking ahead, the structural similarity to many-body Hamiltonians points toward a broader framework in which entanglement witnesses or detection criteria may be systematically derived by combining insights from many-body physics with multipartite entanglement theory.

We believe these additions significantly strengthen the scope and practical relevance of our results, and demonstrate that the underlying insights of our method extend beyond stabilizer states and to other physical settings.

I do, however, have some comments that should be addressed. My main concern is the following. The motivation for minimizing the m -body correlators is that it is hard to perform them. However, to estimate the expectation value of an m -body Pauli string, it suffices to perform single-qubit Pauli measurements and record the parity of all measurements. Single-qubit Pauli measurements are clearly easier to implement than larger weight Pauli strings (this is essentially one of the biggest motivations for studying this problem, according to the authors). Unless I am missing something, this severely limits the motivation for minimizing the weight of Pauli measurements performed.

There are several reasons I could think of why such single-qubit Pauli measurements might be bad, however. Clearly for each measurement one can only estimate the expectation value of Pauli measurements that are locally commutative (i.e. subsets of stabilizers whose restrictions to individual qubits pairwise commute), which would correspond for example to the 'vertex stabilizers' that form independent sets in the graph. After such a measurement, the state would need to be prepared again (note that this also motivates the concept of local measurement settings studied previously). Here I could imagine the authors' approach could provide some gain, since the m -body stabilizers S_i can be measured in succession, without having to recreate the state. At the same time, if i and j are connected, then after measuring S_i and S_j there is no use in measuring $S_i * S_j$, since measuring S_i and S_j projects the state into a well-defined eigenstate of their product. So this would require repreparing the state anyway when measuring the $S_i * S_j$ stabilizers. Furthermore, it is not clear whether performing some of these multi-body measurements in succession (without having to reprepare the state) is actually beneficial in an experiment, since this incurs extra decoherence because the state is idling and one measures larger body correlators.

It might be that the above concern is easily addressed, since I'm not an expert on (experimental considerations) of entanglement witnesses. If it is, it would be good to highlight in the paper what the experimental drawbacks would be for using single-qubit Pauli measurements, as opposed to measuring the full Pauli strings directly/the procedure that the authors propose.

Response: We thank the reviewer for raising this important point. We fully agree that, in principle, expectation values of m -body correlators can be estimated by performing sequential single-qubit Pauli measurements and data post-processing. In some setups, e.g., ion-trap experiments such as Ref. [1], one can easily measure correlators of many qubits (20 in Ref. [1]), exactly in the way described by the reviewer. Moreover, it is true that a large body of the literature on quantum-state characterization focuses on minimizing the number of required measurement settings by grouping commuting observables that can be measured with a common basis setting. This is one of the motivations behind several works we cited, such as [G. Tóth and O. Gühne, Phys. Rev. Lett. **94**, 060501 (2005)] and [Y. Zhou et al., npj Quantum Inf **5**, 83 (2019)].

However, the experimental platforms that motivate our work, most notably microwave photonic generators (e.g., superconducting circuits) and time-bin encoded optical systems (e.g., time-multiplexed photons in the near-visible spectrum generated from spontaneous parametric down-conversion in combination with Sagnac interferometers, or pairs of optical parametric

amplifiers and Mach-Zehnder interferometers), operate under very different constraints. In these settings, **the dominant experimental limitation is not the number of measurement settings but the exponential degradation of usable statistics with the weight of the measured operator.**

One key reason is that the detection efficiency for a single photon (especially for microwave photons), η , is usually much less than 1. As a result, the effective success probability of detecting all m photons needed to measure a weight- m Pauli string scales as η^m . Therefore, even when single-qubit measurements are used, the number of state preparations required to estimate a weight- $O(n)$ observable to a constant precision grows exponentially in the system size n .

To make this quantitative, in the experiment [Nat. Commun. **16**, 5505 (2025)] that inspired this theoretical work, measurements are performed on microwave photons, with a detection efficiency of $\eta \approx 0.25$. As a consequence, estimating weight-4 Pauli operators required around 10 billion repetitions. Although higher-efficiency microwave-photon detectors exist [Phys. Rev. X **8**, 021003 (2018); Nat. Phys. **14**, 546–549 (2018)], they are much harder to calibrate and are not routinely available.

Optical platforms face similar challenges. State-of-the-art optical detection efficiencies are around 90% (see, e.g., [Phys. Rev. Lett. **129**, 050502 (2022)]), but when combined with optical modules allowing fast basis switching between photons in adjacent time bins, the effective efficiency typically drops to around 80% (see, e.g., [Phys. Rev. Lett. **115**, 250401 (2015); Phys. Rev. Lett. **115**, 250402 (2015)]). Furthermore, optical systems often require non-deterministic processes, such as parametric down-conversion and fusion operations for entanglement generation, further amplifying the statistical cost of measuring higher-weight operators. Thus, even though the correlators are accessible via single-qubit measurements, **measuring high-weight operators remains experimentally challenging.**

For these reasons, even though any stabilizer can in principle be estimated using single-qubit measurements, **the dominant experimental cost in these platforms is proportional to η^{-m} .** Therefore, reducing the maximum stabilizer weight from, say, $m = O(n)$ to a constant independent of n can reduce the required number of experimental repetitions by many orders of magnitude.

This is precisely the motivation of our approach: by ensuring that our GME/ k -inseparability criteria involve only stabilizers whose weight is bounded by twice the maximum degree of the underlying graph (constant for many important graph states such as 1D/2D clusters), we dramatically reduce the experimental time needed to estimate them in the photonic platforms of interest.

To clarify these points, we have **added a corresponding explanation to the Introduction (see the revised 4th paragraph).**

Minor concerns:

It took some time to understand what was meant in theorem 1 under eq 4. It would help to be more precise here, and it would also help (but is not necessary) to have Fig. 1 on the same page.

Response: We agree that the notation under Eq. (5) [Eq. (4) in the previous version] in Theorem 1 needs further clarification and may be made more precise. We have now **repositioned Fig. 1 so that it appears on the same page.** Furthermore, we have **substantially revised Fig. 1 and its caption** to better illustrate the corresponding graph-theoretic objects and intuition behind our formulation.

To clarify the conceptual origin of our criteria, we have added explanations around Eq. (2) and Theorem 1 to emphasize why matchings and cuts naturally arise in our bounds. The key idea is that

the entire set of stabilizers is not actually needed to obtain good bounds for certifying entanglement. Once the union of all the edges corresponding to the stabilizers form a connected graph, a polynomially small subset of local stabilizers, together with the positivity of quantum states, is already sufficient to certify important properties of many-body quantum states.

More precisely, the main tools we use are the *anticommutativity inequality* (Lemma 2 and Proposition 1 in Methods) and the anticommuting structure of Pauli operators. These two ingredients jointly yield the analytic bounds in Eqs. (4) and (8). In particular, whenever two qubits a and b in ρ correspond to adjacent vertices in the underlying graph and belong to different groups of a given k -partition of the n qubits, every k -product state of that k -partition necessarily satisfies $|\langle S_a \rangle| + |\langle S_b \rangle| + |\langle S_a S_b \rangle| \leq 1$, $|\langle S_a \rangle| + |\langle S_a S_b \rangle| \leq 1$ and $|\langle S_a \rangle| + |\langle S_b \rangle| \leq 1$. These constraints arise because, after factorization across the partition, the operators acting on each subsystem form anticommuting triples or pairs whose squared expectation values sum to at most one. Alternatively or additionally, one may also derive bounds by leveraging the positivity constraint via a semidefinite programming (SDP) approach, as done in Sec. II C.

To illustrate this more concretely, consider the example of a 5-qubit graph in Fig. 1 of the manuscript, and the partition of the 5 qubits/vertices into three sets, $\{1,4,5\}$, $\{2\}$, and $\{3\}$. Since Eq. (2) contains sums over stabilizers for all vertices and products of stabilizers corresponding to all pairs of neighboring vertices, the quantity in Eq. (2) contains (amongst others) terms such as $|\langle S_1 \rangle| = |\langle X_1 Z_2 Z_3 Z_5 \rangle|$, $|\langle S_2 \rangle| = |\langle X_2 Z_1 Z_3 \rangle|$, and $|\langle S_1 S_2 \rangle| = |\langle Y_1 Y_2 Z_5 \rangle|$, where the fact that vertices 1 and 2 are connected by an edge means that the Pauli- Y terms appear as products of X and Z operators for both qubits 1 and 2. For any state that is separable with respect to the chosen partition in Fig. 1, these stabilizer expectation values factorize as $|\langle S_1 \rangle| = |\langle X_1 Z_5 \rangle| |\langle Z_2 Z_3 \rangle|$, $|\langle S_2 \rangle| = |\langle Z_1 \rangle| |\langle X_2 Z_3 \rangle|$, and $|\langle S_1 S_2 \rangle| = |\langle Y_1 Z_5 \rangle| |\langle Y_2 \rangle|$. Hence, the sum of these three terms can be bounded via the Cauchy-Schwarz inequality,

$$\begin{aligned} |\langle S_1 \rangle| + |\langle S_2 \rangle| + |\langle S_1 S_2 \rangle| &= |\langle X_1 Z_5 \rangle| |\langle Z_2 Z_3 \rangle| + |\langle Z_1 \rangle| |\langle X_2 Z_3 \rangle| + |\langle Y_1 Z_5 \rangle| |\langle Y_2 \rangle| \\ &\leq (|\langle X_1 Z_5 \rangle|^2 + |\langle Z_1 \rangle|^2 + |\langle Y_1 Z_5 \rangle|^2)^{1/2} (|\langle X_2 Z_3 \rangle|^2 + |\langle Z_2 Z_3 \rangle|^2 + |\langle Y_2 \rangle|^2)^{1/2}. \end{aligned}$$

For the sums of squared expectation values under each square root, we can use the anticommutativity bound (Lemma 2 and Proposition 1 in Methods) to bound them by a constant (1 in this case): for each side of the partition, the Pauli operators $\{X, Y, Z\}$ (tensored with spectator Pauli- Z 's) form a maximally anticommuting triple. We can also make similar arguments for any other pair of neighboring vertices that is “separated” by the selected partitioning, and then repeat this procedure for all possible k -partitions, thus arriving at a bound for all k -separable states.

A key structural point is that this argument applies only when the two vertices are connected by an edge and separated by the partition. Thus, by including all vertex stabilizers and all pairs of stabilizers for all edges, we (i) maximize the number of terms that contribute to the left-hand side of the bound, while (ii) ensuring that we still have a maximal set of anticommuting operators under each square root after the factorization.

In contrast, states that are inseparable across this partition can exceed these bounds since the three sums at the beginning can reach a maximum of 3, 2 and 2, respectively (e.g., when $\rho = |G\rangle\langle G|$). These structural dependence on the graph adjacencies, arising specifically from our chosen subset of stabilizers, explain why matchings and cuts naturally enter the formulation of the optimal bounds and, crucially, why our approach can certify not only GME but also general k -inseparability, which most conventional stabilizer-based criteria cannot.

To highlight these points more clearly, we have now also **(i) added a concise explanation after Eq. (2)**, and **(ii) inserted a clarifying remark immediately before Theorem 1**.

It would be good to mention the notion of locally equivalent graph states after remark 1. In particular that one optimize the graph used in the entanglement witness over all graphs in the local equivalence class. For example, all graphs in Fig. A.2 are locally equivalent (if we include qubit permutations). Furthermore, the second bound in eq (3) motivates the study of optimizing the number of edges in graph states up to local complementations, which can be done using the results in for example 2506.00292.

Response: We thank the referee for this helpful suggestion. Following the comment, we have **added a dedicated paragraph after Remark 1** that introduces the notion of LC-equivalent graph states and explains how this freedom can be used to optimize our criteria. In particular, we clarify that one may choose an LC-equivalent representative whose graph has a smaller maximum degree or fewer edges, thereby reducing stabilizer weights or the total number of terms that need to be measured in Eq. (2). This point is now **highlighted in the newly added Remark 2**. We have also **updated the abstract and the introduction (third-to-last paragraph)** accordingly. In addition, we describe how the witness is evaluated using the LC-conjugated stabilizers of the chosen representative state, and we reference the methods of Ref. [76] (arXiv:2506.00292) for identifying minimum-edge representatives.

To better illustrate these ideas, we have **added a new Appendix A.V**, where we give an explicit example showing how local complementations can reduce the maximum degree and the number of edges of a 5-vertex graph. In that appendix, we also explain how to construct a new set of criteria that require fewer stabilizers with a smaller maximum weight.

I think remark 2 could be sharpened, by first saying that the criterion based on G is tailored to $|G\rangle$, then saying that it achieves $n+\gamma|E|$ for $|G\rangle\langle G|$, and then that it can still apply to other non-stabilizer states.

Response: Thank you for the suggestion. We have rewritten this remark (i.e., Remark 3 in the new version) according to your feedback.

Proposition 1 has the definition of the anti-commutator, while it is already used in lemma 2.

Response: Thanks for spotting that. We have now moved the definition of anti-commutator from Proposition 1 to Lemma 2.

I would change the word 'define' in lemma 2 by 'let'.

Response: We agree that this sounds better and have replaced the word “define” with the word “let”.

"As the stabilizer term $|\langle S_b \rangle|$ of $b \in V_{\text{match}}^k$ is* already paired (...)"

Response: Thanks for spotting this grammatical mistake. We already added “is” accordingly.

In A.1 the authors highlight how other witnesses do not detect between different levels of k -inseparability. I feel this could be stressed more in the main text as a novel result.

Response: We agree that this is an important point and merits clearer emphasis in the main text. In particular, our criteria not only detect GME but also distinguish between different levels of k

-inseparability, unlike most existing stabilizer-based witnesses (e.g., Refs. [45, 46, 50]). To make this novel aspect more visible, we have added an explicit remark highlighting this distinction at the end of the paragraph that includes Eq. (3) and discusses the intuition behind our construction. In the following paragraph, we further emphasize this point and reference Appendix A.1 for additional details.

Reviewer #3 (Remarks on code availability):

The code does not include the calculations for the expectation values from the density matrices, as far as I can tell.

Response: We thank the referee for the comment. Indeed, initially the code freely available on the referred GitHub repository only contained the notebooks used to obtain the simulated noisy states. We have now added the notebooks used for the computation of the expectation values. In the process, we reorganized the contents of the repository and extended the readme file to clarify the function and expected usage of each of the notebooks.

Response to Reviewer 1

Dear Referee,

Thank you again for reviewing our manuscript for *Nature Communications*. We appreciate the time and effort that you put into the review process and providing constructive feedback on our manuscript. Please find our in-line response below.

With best regards,

Nicky Kai Hong Li (on behalf of all authors)

Inline Response:

Reviewer #1 (Remarks to the Author):

The concerns and issues raised in the previous round of review have been carefully and adequately addressed by the authors in the revised manuscript. The current version demonstrates improved clarity and correctness. Therefore, I recommend this work for publication.

Response: We sincerely thank the referee again for the thorough and encouraging assessment of our work, and for all the constructive comments that have improved the clarity of the manuscript.

Response to Reviewer 2

Dear Referee,

Thank you again for reviewing our manuscript for *Nature Communications*. We appreciate the time and effort that you put into the review process and providing constructive feedback on our manuscript. Please find our in-line response below.

With best regards,

Nicky Kai Hong Li (on behalf of all authors)

Inline Response:

Reviewer #2 (Remarks to the Author):

The authors addressed my major and minor concerns. It reads much better now with a clear explanation of the key concept.

However, due to the somewhat incremental nature (using locally commuting stabilizer operators with Cauchy Schwarz), I am not 100% sure whether the paper represents an advance of significance to specialists within the field.

Response: We thank the referee for providing useful feedback on the manuscript and for acknowledging that the major and minor concerns have been addressed, leading to improved clarity and presentation.

We note the referee's comment regarding the perceived incremental nature of the approach. As clarified in the manuscript, our aim is to provide a practically motivated and experimentally accessible framework that significantly reduces measurement overhead for multipartite entanglement detection under realistic constraints. We believe this perspective, together with the generality of the framework and its applicability to current multi-qubit platforms, will be of interest to specialists in the field.

Response to Reviewer 3

Dear Referee,

Thank you again for reviewing our manuscript for *Nature Communications*. We appreciate the time and effort that you put into the review process and providing constructive feedback on our manuscript. Please find our in-line response below.

With best regards,

Nicky Kai Hong Li (on behalf of all authors)

Inline Response:

Reviewer #3 (Remarks to the Author):

The authors have addressed all my questions, in particular my concern regarding minimizing the weight of the stabilizers is sufficiently addressed. As such, I now recommend acceptance in *Nature Communications*.

Response: We sincerely thank the referee again for the thorough and encouraging assessment of our work, and for all the constructive comments, in particular regarding minimizing stabiliser weight with local complementations, which have improved the clarity of the manuscript and the applicability of our results.